# Network-wide thermodynamic constraints shape NAD(P)H cofactor specificity of biochemical reactions

Pavlos Stephanos Bekiaris[1] & Steffen Klamt ⬡[1] ✉

The ubiquitous coexistence of the redox cofactors NADH and NADPH is widely considered to facilitate an efficient operation of cellular redox metabolism. However, it remains unclear what shapes the NAD(P)H specificity of specific redox reactions. Here, we present a computational framework to analyze the effect of redox cofactor swaps on the maximal thermodynamic potential of a metabolic network and use it to investigate key aspects of redox cofactor redundancy in *Escherichia coli*. As one major result, our analysis suggests that evolved NAD(P)H specificities are largely shaped by metabolic network structure and associated thermodynamic constraints enabling thermodynamic driving forces that are close or even identical to the theoretical optimum and significantly higher compared to random specificities. Furthermore, while redundancy of NAD(P)H is clearly beneficial for thermodynamic driving forces, a third redox cofactor would require a low standard redox potential to be advantageous. Our approach also predicts trends of redox-cofactor concentration ratios and could facilitate the design of optimal redox cofactor specificities.

The redox cofactors NAD (nicotinamide adenine dinucleotide) and NADP (nicotinamide adenine dinucleotide phosphate), which differ only in a phosphate group, play an essential role as electron carriers in the metabolism of all types of living cells. Both cofactors (or coenzymes) occur either in the oxidized form (NAD$^+$ and NADP$^+$), which can take up two electrons in oxidation reactions, or in the reduced form (NADH and NADPH), which functions as electron donor enabling the reduction of metabolites. Recent work suggests that both cofactors were already present in the last bacterial common ancestor[1] and the specificity of functionally related metabolic enzymes for NAD(H) or NADP(H) is largely conserved in different organisms.

The universal co-occurrence of NAD(H) and NADP(H) raises the question why there are two different pools of redox cofactors with very similar chemical properties. It is a common view that the existence of two pools enables the parallel operation of metabolic pathways with different requirements in the thermodynamic potentials of the used redox cofactor. While the *standard* Gibbs free energy changes between oxidized and reduced forms of NAD(H) and NADP(H) (and thus their redox potentials) are nearly identical, the actual Gibbs free energies differ largely in vivo. This is because the in vivo ratio of reduced and oxidized form is typically very low for NADH/NAD$^+$ (e.g. -0.02 in *Escherichia coli*), but very high for NADPH/NADP$^+$ (-30 in *E. coli*)[2]. This enables simultaneous operation of oxidation reactions (through a low NADH/NAD$^+$ ratio) and reduction reactions (through a high NADPH/NADP$^+$ ratio), which might be impossible with a single cofactor pool. Consistent with this view, NAD$^+$ predominantly functions as electron acceptor in catabolic reactions, while NADPH typically acts as electron donor in biosynthetic pathways. However, a simple association of NAD(H) with catabolism and NADP(H) with anabolism neglects the fact that the consumed NAD$^+$ and NADPH must be recycled again by appropriate reactions. For example, in heterotrophic organisms, on which we will focus herein, NAD$^+$ is mainly recycled by respiration and fermentation pathways. Conversely, the NADPH pool is often, to a large extent, replenished via the oxidative pentose phosphate pathway, which is itself a catabolic route.

[1]Max Planck Institute for Dynamics of Complex Technical Systems, Sandtorstr. 1, Magdeburg, Germany. ✉e-mail: klamt@mpi-magdeburg.mpg.de

While there is a common view on the functional role of the two coenzymes, unbiased quantitative measures are needed that can explain why redundant redox cofactor pools bring an evolutionary advantage for the cell. An important contribution in this direction has been made in a recent work by Goldford et al.[3] The authors used constraint-based modeling to analyze, for example, thermodynamic feasibility of growth in a metabolic network under single cofactor swaps as well as kinetic simulations in a small example network to illustrate that the co-existence of multiple, functionally associated cofactor pools like NAD(H) and NADP(H) may reduce the total amount of enzymes needed to catalyze metabolic fluxes due to increased thermodynamic driving forces. However, there are still several open questions, in particular, what shapes the cofactor specificity of the metabolic reactions (and associated enzymes) in a given metabolic network? For example, due to the low NADH/NAD$^+$ ratio in the cell, the driving force of a reaction reducing one of the two cofactors would always benefit if it used NAD$^+$, however, as already mentioned above, at least some reactions must use NADP$^+$ to replenish the NADPH pool. The optimal distribution of NAD(P)(H) specificities becomes thus a problem at the network level. If we were able to freely assign cofactor specificities to all reactions, which distribution would be optimal (e.g., in terms of thermodynamic driving force in the network) and could we then even predict the required NAD(P)H/NAD(P)$^+$ ratios without fixing them to predefined (known) values?

In this work, in order to address those and other questions and to gain insights related to cofactor redundancy, we introduce a framework called TCOSA (Thermodynamics-based COfactor Swapping Analysis), which enables us to analyze the effect of redox cofactor swaps on the thermodynamic potential of a given genome-scale metabolic network. Similar to Goldford et al.[3], TCOSA relies on constraint-based metabolic modeling in combination with thermodynamic constraints (standard Gibbs free energies and metabolite concentration ranges) but, as a key difference, it uses the notion of the max−min driving force (MDF)[4,5] to assess the maximal thermodynamic driving force achievable in the network. This approach does not need kinetic parameters and allows the calculation of various properties related to thermodynamic effects of cofactor swaps. In particular, it enables us to predict NAD(P)(H) specificities in metabolic reactions that maximize the overall thermodynamic driving force, which can then be compared with the wild-type specificities. We use TCOSA to analyze the thermodynamic driving forces of different scenarios of cofactor specificities in a genome-scale metabolic network of *E. coli*. We find that the wild-type NAD(P)(H) specificities of the metabolic reactions in *E. coli* enable in almost all cases maximal or close to maximal thermodynamic driving forces and are thus, to a large extent, governed by network structure and thermodynamics alone. Among several other important results, we find that providing more than two redox cofactor pools does not significantly increase the maximal thermodynamic driving forces unless the redox potential of the third redox couple is different from that of NAD(P)H. We also discuss possible applications of the TCOSA framework as a design tool for metabolic engineering, e.g. to increase thermodynamic driving forces for synthesis of a target product.

## Results

### Model preparation and NAD(P)(H) specificity scenarios

Our developed TCOSA framework (described in detail in the Methods section) facilitates a systematic analysis of the effects of altered NAD(P)(H) specificities of redox reactions on the achievable thermodynamic driving forces in a given metabolic network. Herein we applied this framework to *i*ML1515[6], the latest genome-scale metabolic model of *E. coli*, which we initially reconfigured to prime it for TCOSA-related calculations (see Methods). Essentially, each NAD(H)- and NADP(H)-containing reaction is duplicated but with the alternative cofactor (see Methods and Fig. 1a).

As detailed below, this reconfigured model (called *i*ML1515_TCOSA) is used in several calculations to analyze effects of different variations of cofactor swaps. Basically, we consider four distinct scenarios of NAD(P)(H) specificities in the network (Fig. 1b):

1. *Wild-type specificity*: This scenario assumes the original NAD(P)(H) specificity of the *i*ML1515 model. Hence, if a reaction originally utilized NAD(H), then its NADP(H) variant is blocked in the *i*ML1515_TCOSA model (reaction flux fixed to 0). Conversely, if a reaction originally used NADP(H), its NAD(H) variant is blocked.

2. *Single cofactor pool*: All NADP(H) variants are blocked, hence, all redox-cofactor-dependent reactions use NAD(H). To keep the growth reaction feasible, the amount of NADP$^+$ consumed by this reaction is provided from the NAD$^+$ pool.

3. *Flexible specificity*: For all reactions consuming redox cofactors, both variants (NAD(H) or NADP(H)) are available. Hence, in this scenario, any optimization procedure can freely choose between NAD(H) or NADP(H) dependency to maximize its objective function. However, constraints are used to ensure that either the NAD(H) or the NADP(H) variant (but not both) of a reaction can be active at the same time (see Methods).

4. *Random specificity*: Here, through a simulated stochastic coin flip, either the NAD(H) or the NADP(H) variant of a reaction is active (and the other is blocked), regardless of its original state in *i*ML1515. In all random simulations performed below, a total of 1000 random specificity distributions was generated and analyzed, of which 500 have a free and 500 a fixed pool size. In the fixed pool size, it is ensured that the number of active NAD(H) and of active NADP(H) reactions equals the respective original numbers in *i*ML1515. Any random specificity that resulted in thermodynamic infeasibility or near-infeasibility (MDF < 0.1 kJ/mol; see Methods) was disregarded from subsequent analyses.

The calculations were performed for growth on glucose (and later also on acetate) under aerobic as well as anaerobic conditions (oxygen uptake blocked). Further details on the model can be found in the Methods section.

Using flux balance analysis[7], initially without consideration of thermodynamic constraints, we determined the maximal growth rate of *i*ML1515_TCOSA with the wild type and single-cofactor specificities. For the wild type, we found $\mu_{max} = 0.877\,h^{-1}$ for aerobic and $\mu_{max} = 0.375\,h^{-1}$ for anaerobic conditions, which, as expected, match exactly the values from the original *i*ML1515 model. Interestingly, the maximal growth rates for the single-cofactor scenario are slightly higher for aerobic (0.881 h$^{-1}$) and even significantly larger for anaerobic conditions (0.470 h$^{-1}$), indicating that allowing usage of NAD(H) in all reactions utilizing NAD(P)H is stoichiometrically more efficient for growth (but thermodynamically likely infeasible). In all subsequent calculations where maximum growth is demanded, we used 99% of the respective maximum value to avoid numerical issues.

### Comparing driving forces for different specificity scenarios

Next, we used the *i*ML1515_TCOSA model to compare maximal thermodynamic driving forces achievable with the different cofactor specificity scenarios. As a global measure for the network-wide thermodynamic potential we used the notion of max−min driving force. Generally, driving forces can be defined at different levels (Fig. 2). The driving force of a single reaction is the negative Gibbs free energy change ($-\triangle_r G'$) of this reaction, the driving force of a pathway is the minimum of all driving forces of the reactions involved, and, according to Noor et al.[4], the max−min driving force (MDF) of a given pathway is the maximal possible pathway driving force (within given bounds for metabolite concentrations; Fig. 2). While this original MDF definition is useful to quantify the maximal driving force of a given pathway, a related approach[5] (called OptMDFpathway) goes one step further: it starts with an entire metabolic network and a

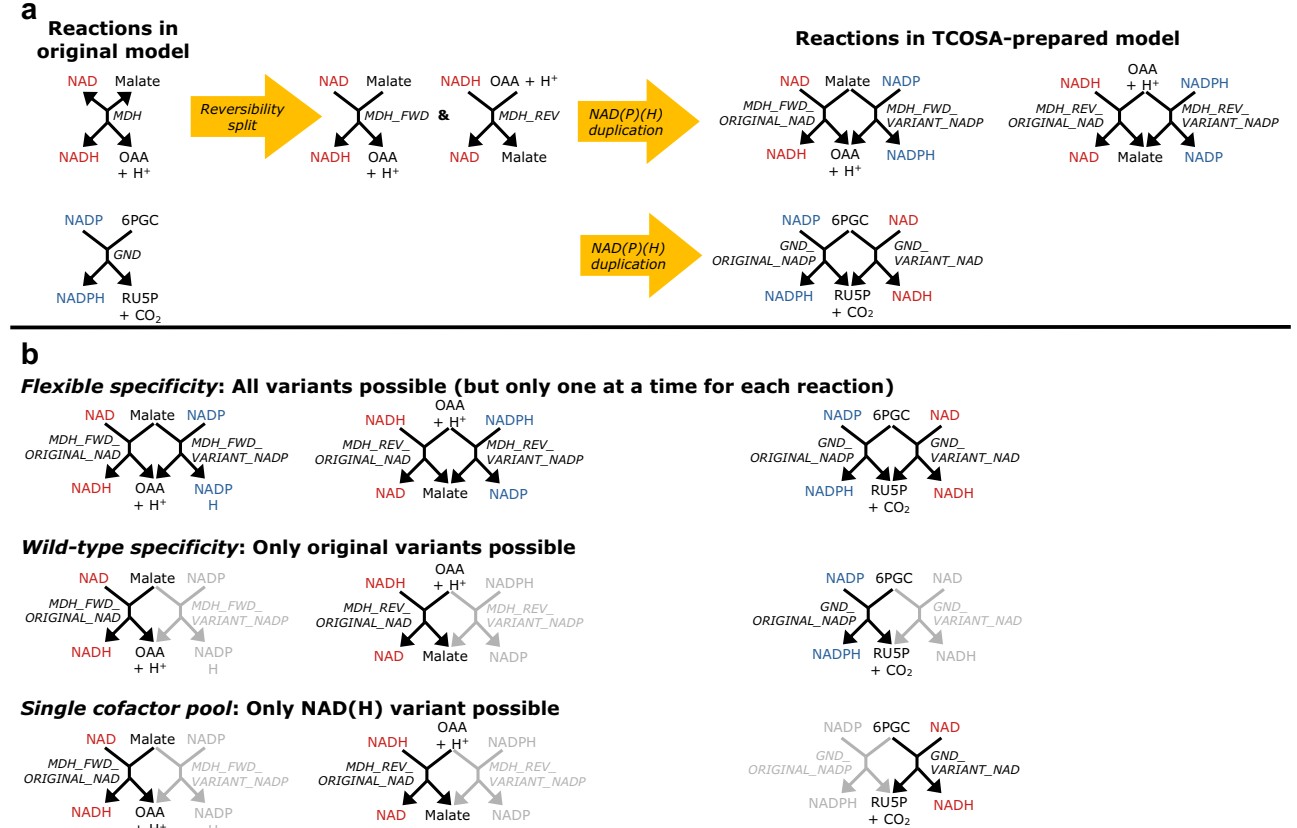

**Fig. 1 | TCOSA model reconfigurations and definition of specificity scenarios.**
**a** Examples of TCOSA model reconfigurations to enable different redox cofactor
specificities for NAD(P)H-dependent reactions (MDH: malate dehydrogenase
reaction; GND: phosphogluconate dehydrogenase reaction; OAA: oxaloacetate;
6PGC: 6-phosphogluconate; RU5P: ribulose-5-phosphate). **b** Different scenarios of
cofactor specificities analyzed in this work. The generation of random specificities
looks similar to the wild-type specificity, with the difference that the choice of
NAD(H) or NADP(H) specificity is random.

demanded phenotypic behavior (e.g., a given growth rate) and sear-
ches then for a steady-state flux distribution and suitable metabolite
concentrations that maximize the MDF for the predefined phenotypic
behavior (note that all *active* reactions of the found flux distribution
have a driving force that is at least as high as the found MDF; Fig. 2). In
the following, we will always refer to this generalized MDF definition,
i.e., when considering the MDF for a given condition, we mean the
maximal achievable MDF together with the associated fluxes and
metabolite concentrations. All these values can be computed via a
mixed-integer linear program (see Methods). Generally, the rationale
to analyze network-wide thermodynamic driving forces is based on the
fact that low driving forces for certain reactions (indicated by a low
MDF) will limit their flux or must be compensated by very large enzyme
demands and thus enzyme costs[8]. Hence, by computing and compar-
ing the MDF for the different cofactor specificities we can assess the
potential of thermodynamic limitations associated with the different
specificities.

We computed and compared the MDF values for each of the four
NAD(P)H specificity scenarios under aerobic as well as anaerobic
conditions and for different growth rates, starting from the respective
maximal growth rate and then reducing it in steps of $0.05\,h^{-1}$ until a
growth rate of less than $0.05\,h^{-1}$ would be reached ($0.05\,h^{-1}$ is the
smallest growth rate considered). This discretization results in 18
analyzed growth rates for aerobic conditions ($0.868\,h^{-1} \dots 0.05\,h^{-1}$)
and 8 for anaerobic conditions ($0.371\,h^{-1} \dots 0.05\,h^{-1}$).

The results are summarized in Fig. 3. We started with the flexible
specificity scenario (free choice of NAD(P)H specificity) and found a
maximal MDF of ca. 8 kJ/mol for aerobic (Fig. 3a) as well as for anaerobic
(Fig. 3c) conditions. In the aerobic case, this maximum MDF value is

reached for almost all growth rates in this scenario; only for growth rates
close to the maximum it is slightly reduced. Under anaerobic conditions,
the MDF value of the flexible scenario is significantly reduced for high
growth rates (less than 50%) and approaches the maximum value only
for the smallest growth rates. Generally, it is obvious that, for a given
growth rate, the MDF value obtained with the fully flexible scenario is the
highest that can be reached in all scenarios since it is the least con-
strained scenario and is even allowed to change cofactor specificities for
different growth rates and conditions (which indeed happens). Impor-
tantly, looking at the MDF achievable with wild-type specificity, we
notice that it reaches, for smaller and medium growth rates, exactly the
MDF value of the flexible specificity. Only for growth rates that are closer
to the maximum a small to moderate difference can be observed, which
is at most pronounced under anaerobic conditions within a smaller
window of growth rates between $0.25\,h^{-1}$ and $0.3\,h^{-1}$. Supplementary
Data 1 presents a summary of the properties of two selected MDF-
optimal solutions (one for a high growth rate of aerobic and one for a
high growth rate of anaerobic conditions), including a variability analysis
of fluxes and their thermodynamic potentials. We also computed the
feasible ranges of metabolite concentrations for the MDF-optimal solu-
tions and found that almost all of these ranges are compatible with
measured metabolite concentrations provided in Bennett et al.[2].

In the single cofactor pool scenario, where all reactions use
NAD(H) as redox cofactor, the MDF reaches, with a single exception,
for all growth rates under both aerobic and anaerobic conditions sig-
nificantly smaller values than the flexible and the wild-type scenario.
This already indicates that the existence of two cofactor pools
enhances thermodynamic driving forces in the wild-type scenario.
Furthermore, the MDF values of the 1000 random specificities

Example network with (given) standard Gibbs free energies of reactions:

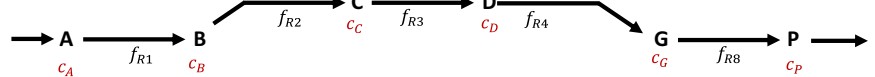

Driving force of a single reaction: $f_i = -\Delta_r G'_i = -\Delta_r G'^\circ_i - R \cdot T \cdot \sum_j N_{ji} \cdot \ln(c_j)$

Example: $R_1 (A \rightarrow B)$ with fixed concentrations: $f_{R1} = -\Delta_r G'^\circ_{R1} - R \cdot T \cdot (-\ln(c_A) + \ln(c_B))$

**Driving force of a specific pathway**: minimum driving force of all reactions of the pathway

$f_{Pathway} = \min(f_{R1}, f_{R2}, f_{R3}, f_{R4}, f_{R8})$ with fixed concentrations

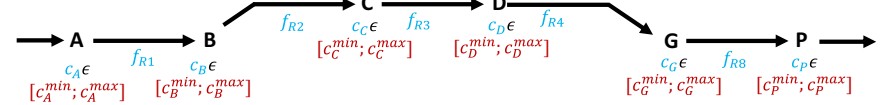

**Max-min driving force (MDF) of a pathway**: maximal achievable driving force of the pathway

$MDF_{Pathway} = \max(f_{Pathway}) = \max(\min(f_{R1}, f_{R2}, f_{R3}, f_{R4}, f_{R8}))$ within given concentration ranges

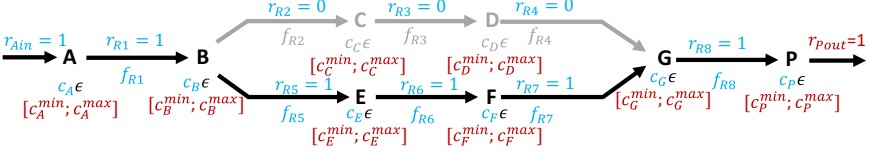

**MDF of network** (also called OptMDF): find flux distribution **r** maximizing MDF of active reactions

$MDF_{Network} = \max(MDF(\mathbf{r}))$ with given concentration ranges and flux constraints

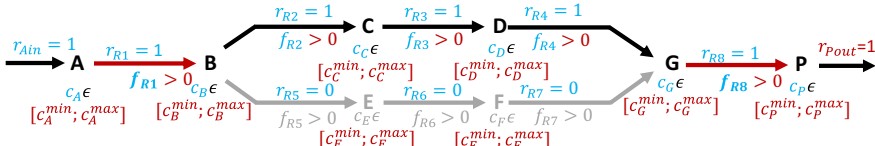

**MDF of subnetwork (SubMDF)**: find flux distribution **r** maximizing MDF of selected set of reactions

e.g., SubMDF = MDF$(R_1, R_8)$ with given concentration ranges and flux constraints
and demanded minimal MDF (e.g., >0 kJ/mol) of other active reactions

**Fig. 2 | Overview of different notions of driving forces and max−min driving forces (MDF) relevant in this paper illustrated by a small example network.** Red color indicates (given) constraints and blue color returned results of the calculation (or optimization) routine. For precise formulations of the concepts see Methods.

demonstrate that the wild-type distribution of NAD(P)(H) specificities behave significantly better than a random choice. Depending on the growth rate, at most 0.1% (aerobic conditions) or 7.9% (anaerobic conditions) of the 1000 tested random NAD(P)(H) specificities have a better MDF than the wild-type specificity (Fig. 3a, c, Table 1). Depending on the conditions and growth rate, some more random specificities reach the same MDF value as the one of the wild type, however, in all cases at least 83% of the random specificities perform worse. Notably, at some growth rates, there are also random specificities whose maximal thermodynamic driving force is even worse than the one with a single cofactor pool. This can happen, for example, if in a random NAD(P)(H) specificity only very few and thermodynamically unfavorable pathways exist for synthesizing the required amounts of NADPH.

By definition, the MDF demands that the driving forces of all active reactions of a flux distribution are at least as high as the MDF. We here introduce the relaxed notion of SubMDF (see Methods and Fig. 2): it demands that only a selected subset of all active reactions

in the flux distribution (in our application the active NAD(P)(H)-depending reactions) must reach the respective MDF value, while for all other reactions it is only demanded that their driving force is above 0.1 kJ/mol. In this way, we can exclude thermodynamic bottlenecks from our considerations that have no relation to NAD(P)(H)-using reactions and focus our analysis instead on the effects of varying NAD(P)(H) specificities on thermodynamic driving forces in redox reactions. As expected, the SubMDF values are generally higher in the different scenarios than the corresponding MDF values, since now only the driving forces of a subset of the active reactions need to reach the MDF value (Fig. 3b, d; Table 1). While the SubMDF results look qualitatively similar to the MDF version, the absolute differences between the different NAD(P)(H) specificity scenarios are partially more pronounced. For maximal growth rates, the relative gap between the SubMDF of the wild type vs. the flexible specificity is now somewhat larger. For aerobic conditions, we see again that, with smaller growth rates, this difference shrinks until the wild-type specificity reaches the theoretical optimum of the

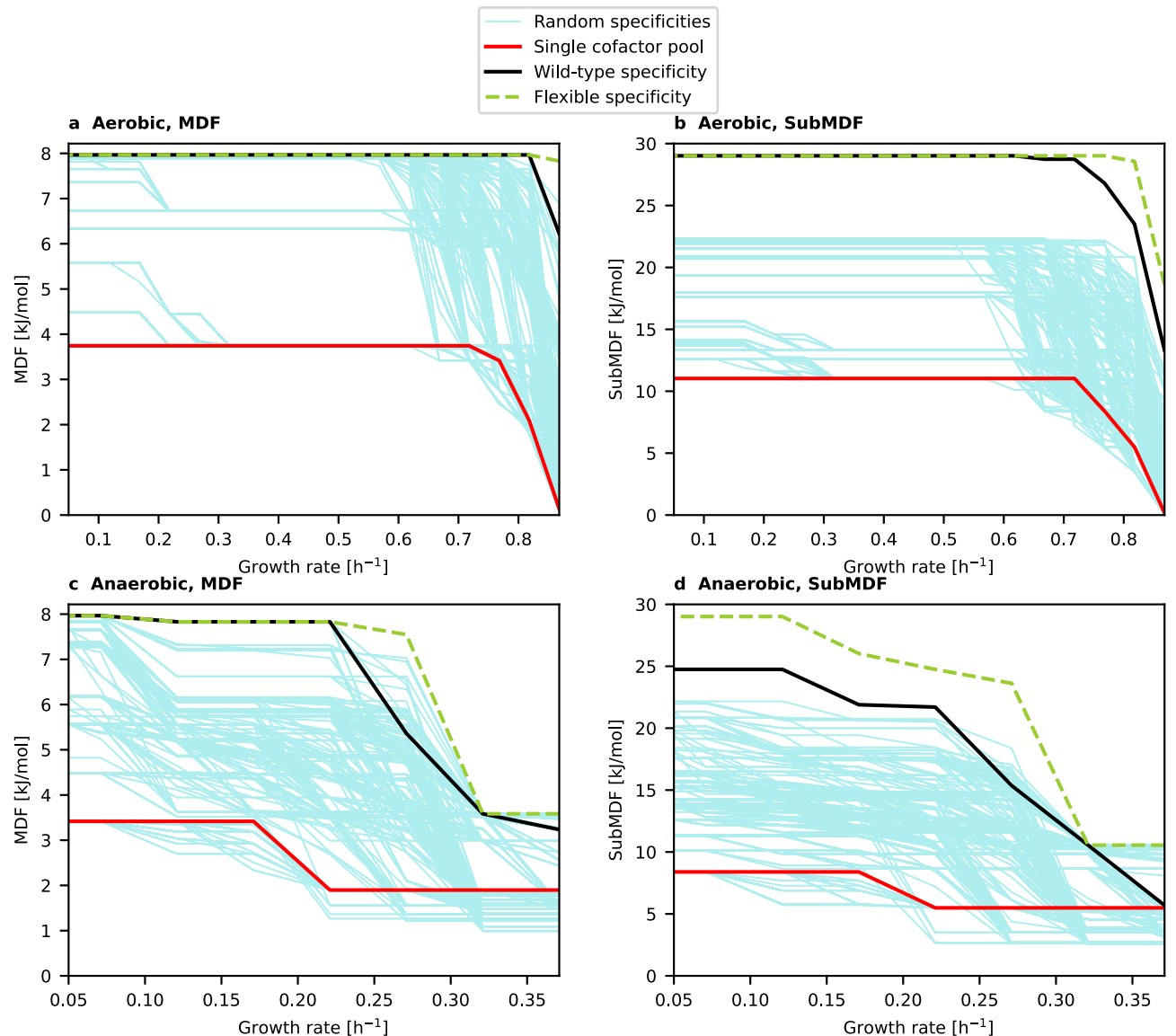

**Fig. 3 | Maximal thermodynamic driving forces (MDF and SubMDF) achievable with different NAD(P)(H) specificities. a** Aerobic conditions with MDF as optimization target. **b** Aerobic conditions with SubMDF as target. **c** Anaerobic conditions with MDF as target. **d** Anaerobic conditions with SubMDF as target. Source data are provided as a Source Data file.

flexible specificity. Moreover, not a single random specificity reaches the SubMDF of the wild-type specificity for any of the growth rates. For anaerobic conditions, the SubMDF value of the wild type coincides with the flexible scenario for one growth rate while a moderate difference remains for the other growth rates. For the maximum growth rate, 13.8% of the random specificities perform better or equal than the one of the wild type but for medium or small growth rates all random specificities have a lower SubMDF value compared to the wild type. Generally, for small and medium growth rates, the relative gap between the SubMDF values of the flexible/wild-type specificity scenarios on the one side and the best random and single cofactor pool scenarios on the other side is larger than for the MDF values.

Taken together, the results presented in Fig. 3 and Table 1 clearly indicate that (a) the network with two cofactor pools reaches much higher potential driving forces than with a single cofactor pool and that (b) the cofactor specificities of the wild type enable driving forces that either coincide or are relatively close to the optimum (represented by the flexible scenario) and are with very high probability larger than randomly chosen specificities.

**Cofactor swaps needed in the wild type to reach maximal MDF**
To analyze how far the wild-type specificity differs from the thermodynamically optimal (flexible) specificity, we determined the number of necessary cofactor swaps in the wild-type specificity model needed to reach the maximal (Sub)MDF values of the flexible scenario (see Methods). Over all tested conditions and growth rates, we found that in 47 out of the 52 considered cases only three or less reaction cofactor swaps are necessary in the wild type to reach the respective (Sub)MDF of the flexible scenario (Table 1). Moreover, consistent with Fig. 3, in 38 cases even no swap was required because the (Sub)MDF value of the wild type reaches the one of the flexible specificities. A singularity is observed for anaerobic growth at growth rate $0.271\,h^{-1}$ where larger numbers (9 and 14, respectively) of swaps are required to reach the theoretically maximal MDF and SubMDF value of the flexible scenario.

Several cofactor swaps occurred in higher frequencies. One example is the $NAD^+$-dependent pyruvate dehydrogenase (PDH) with the reaction formula

$$CoA + pyruvate + NAD^+ \rightarrow acetyl-CoA + CO_2 + NADH.$$

**Table 1 | Relative (Sub)MDF performance of random specificities compared to the wild-type specificity and number of necessary cofactor swaps (starting from the wild-type specificity) to reach the theoretically optimal (Sub)MDF obtained with the flexible specificity**

| Oxygen availability | Growth rate [h⁻¹] | Percentage of random specificities with higher/equal/lower (Sub)MDF compared to the wild-type specificity | | Number of necessary swaps in wild type to reach (Sub)MDF of flexible specificity | |
|---|---|---|---|---|---|
| | | MDF | SubMDF | MDF | SubMDF |
| Aerobic | 0.868 | 0.1%/0%/99.9% | 0%/0%/100% | 6 | 2 |
| Aerobic | 0.818 | 0%/0.4%/ 99.6% | 0%/0%/100% | 0 | 3 |
| Aerobic | 0.768 | 0%/2.3%/ 97.7% | 0%/0%/100% | 0 | 2 |
| Aerobic | 0.718 | 0%/5.6%/ 94.4% | 0%/0%/100% | 0 | 2 |
| Aerobic | 0.668 | 0%/11.8%/ 88.2% | 0%/0%/100% | 0 | 0 |
| Aerobic | 0.618 | 0%/14.1%/ 85.9% | 0%/0%/100% | 0 | 0 |
| Aerobic | 0.568...0.518 | 0%/14.9%/ 85.1% | 0%/0%/100% | 0 | 0 |
| Aerobic | 0.468...0.118 | 0%/15.0%/ 85.0% | 0%/0%/100% | 0 | 0 |
| Aerobic | 0.068 | 0%/17.0%/ 83.0% | 0%/0%/100% | 0 | 0 |
| Aerobic | 0.05 | 0%/17.0%/ 83.0% | 0%/0%/100% | 0 | 0 |
| Anaerobic | 0.371 | 7.9%/0.3%/ 91.8% | 13.4%/0.4%/86.2% | 1 | 1 |
| Anaerobic | 0.321 | 0%/8.7%/ 91.3% | 0%/7.1%/92.9% | 0 | 0 |
| Anaerobic | 0.271 | 3.7%/0.4%/ 95.9% | 1.8%/0.3%/97.9% | 9 | 14 |
| Anaerobic | 0.221 | 0%/0.6%/ 99.4% | 0%/0%/100% | 0 | 2 |
| Anaerobic | 0.171 | 0%/0.6%/ 99.4% | 0%/0%/100% | 0 | 5 |
| Anaerobic | 0.121 | 0%/0.6%/ 99.4% | 0%/0%/100% | 0 | 4 |
| Anaerobic | 0.071 | 0%/2.3%/ 97.7% | 0%/0%/100% | 0 | 3 |
| Anaerobic | 0.05 | 0%/2.3%/ 97.7% | 0%/0%/100% | 0 | 3 |

The total number of random specificities tested was 1000.

At first sight, switching from NAD⁺ to NADP⁺ specificity will most likely lead to a reduced driving force of this single reaction as the NADH/NAD⁺ ratio adjusted by the model is typically much smaller than NADPH/NADP⁺ (reflecting the in vivo situation; see also below). However, in the considered cases, this has no negative effect on the (Sub)MDF, which maximizes the minimal driving forces of all involved reactions. The $\triangle_r G'^{\circ}$ of PDH is markedly negative (−34.37 kJ/mol) allowing for a thermodynamically less favorable cofactor usage of NADP⁺ without becoming a bottleneck for the MDF. In this way, a new route for the thermodynamically efficient production of NADPH becomes possible which in turn abolishes the necessity to use an NADPH-producing reaction with less MDF-favorable thermodynamic parameters eventually improving the MDF. Hence, this cofactor change cannot be explained at the reaction but at the network level.

Another example for a frequently found cofactor swap is the NADP⁺-dependent isocitrate dehydrogenase catalyzing the reaction

$$\text{isocitrate} + \text{NADP}^+ \rightarrow 2-\text{oxoglutarate} + CO_2 + \text{NADPH}.$$

A straightforward explanation for the suggested swap is that switching to NAD⁺ is thermodynamically more favorable due to the established low NADH/NAD⁺ ratio, which helps to overcome the positive $\triangle_r G'^{\circ}$ (+5.12 kJ/mol) of this reaction. Interestingly, NAD-dependent isocitrate dehydrogenases are well-known from other organisms and are assumed to be the ancestor of bacterial NADP⁺-dependent dehydrogenases. However, in E. coli, experimental studies showed that the use of NADPH seems to be especially favorable with acetate as sole substrate, since a switch from NADP(H) to NAD(H) dependency impairs growth under these conditions[9] (see also below). This shows that some wild-type NAD(P)(H) specificities, which appear unfavorable in our study, could become explainable under other growth regimes. A potential mechanism for the cell to adapt the cofactor specificity of a reaction to varying growth conditions would be to use enzymes with different NAD(P)(H) specificities and to adjust their expression depending on the conditions. In the iML1515 model, we found 33 reaction pairs that can operate with both types of redox cofactors, of which 22 are promiscuous activities of the same enzyme and only 11 are catalyzed by isoforms with different cofactor specificity. However, there is only one such reaction pair in the central carbon metabolism, which is catalyzed by the malic enzyme decarboxylating malate to pyruvate. E. coli possesses two variants of the malic enzyme (SfcA and MaeB), one (SfcA) preferring NAD⁺ and one (MaeB) working exclusively with NADP⁺, which are differently regulated[10]. Hence, the mechanism of switching between both cofactor specificities via isoforms of enzymes seems to be of lower relevance in E. coli, at least for central metabolic pathways.

**Effects of single cofactor swaps**

To further analyze how good the wild-type specificities are in terms of network-wide thermodynamic driving forces we introduced, separately for each NAD(P)(H)-dependent reaction, a cofactor swap such that the original reaction is deactivated while the reaction variant with the alternative cofactor is activated, and tested then the effect of this swap on the achievable (Sub)MDF values at the evaluated growth rates. Swaps resulting in a stoichiometrically infeasible solution were disregarded in further calculations.

For a better interpretation of the results, from the set of 238 NAD(P)(H)-dependent reactions we first determined the number of reactions that are involved (active) in any of the (Sub)MDF-optimal flux distributions under any of the investigated conditions (here we also accounted for multiplicity of solutions). We found that 161 of the 238 reactions can contribute to at least one (Sub)MDF-related solution under wild-type specificity. Next, we calculated the number of reactions that had an effect on the (Sub)MDF for at least one of the conditions, if their cofactor specificity is swapped, and found 71 such reactions (44% of the 161 reactions). Table 2 shows the breakdown (and sign) of the effects with respect to (an)aerobiosis and chosen MDF value. It turns out that the great majority of

**Table 2 | Number of reactions (and their percentage with respect to all 161 *i*ML1515 reactions involved in any (Sub)MDF-optimal solution), where a single cofactor swap affects the (Sub)MDF at any of the tested growth rates and conditions. The maximal and average (Sub)MDF increases and decreases are also provided for each condition**

| Oxygenavailability | Number of reactions for which a cofactor swap influences the (Sub)MDF of the wild-type NAD(P)(H) specificity | | | |
|---|---|---|---|---|
| | **MDF** | | **SubMDF** | |
| | Increase | Decrease | Increase | Decrease |
| Aerobic | 1 (0.6%) 0.3 kJ/mol | 16 (9.9%) up to −5.9 kJ/mol, average: −2.6 kJ/mol | 7 (4.3%) up to +3.3 kJ/mol, average: +1.1 kJ/mol | 51 (31.7%) up to −18.4 kJ/mol, average: −6.5 kJ/mol |
| Anaerobic | 3 (1.9%) up to +1.6 kJ/mol, average: +0.6 kJ/mol | 25 (15.5%) up to −5.9 kJ/mol, average: −1.7 kJ/mol | 9 (5.6%) up to +4.9 kJ/mol, average: +1.7 kJ/mol | 36 (22.4%) up to −16.3 kJ/mol, average: −3.6 kJ/mol |

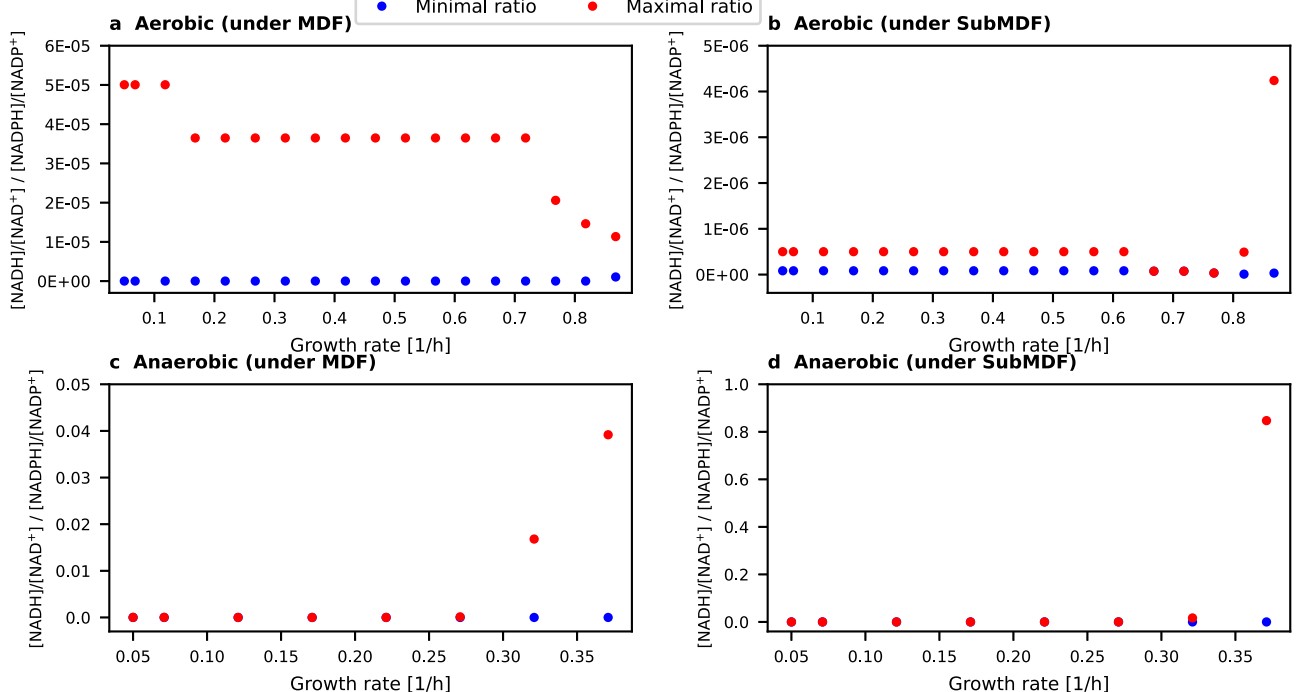

**Fig. 4 | Maximal and minimal possible $\frac{[NADH]/[NAD^+]}{[NADPH]/[NADP^+]}$ ratio to reach the respective growth-rate-associated (Sub)MDF value (for wild-type specificity).**
**a** Aerobic conditions under MDF. **b** Aerobic conditions under SubMDF.
**c** Anaerobic conditions under MDF. **d** Anaerobic conditions under SubMDF. Note that if a blue dot (minimal ratio) is not seen then it largely overlaps with the red dot (maximal ratio) and is thus very close to this value. In all these cases, the maximum value of the $\frac{[NADH]/[NAD^+]}{[NADPH]/[NADP^+]}$ ratio is below 0.001. Source data are provided as a Source Data file.

cofactor swaps having an effect on the (Sub)MDF lead to a decrease of the latter indicating that the wild-type NAD(P)(H) specificities are beneficial with respect to the overall thermodynamic driving force and some single cofactor swaps may profoundly reduce the MDF (up to −5.9 kJ/mol) or SubMDF (up to −18.4 kJ/mol; Table 2). Even though few single cofactor swaps in the wild-type specificity may partially have positive effects on achievable (Sub)MDF values, a single cofactor swap is, in most cases, not enough to reach the optimal (Sub)MDF of the flexible specificity as already shown in the previous section (Table 1).

**Trends of the redox state of cofactors can be predicted**
Next, we investigated whether our model can also be used to predict known trends regarding the degree of reduction of the NAD(H) and the NADP(H) pool. Given that the NAD(H) pool in *E. coli* is usually much more oxidized than the NADP(H) pool[2], one would expect that the model with wild-type specificity predicts a lower [NADH]/[NAD⁺] ratio compared to [NADPH]/[NADP⁺]. To verify this, we computed the possible ranges of the quotient of the two ratios (i.e. the feasible range of $\frac{[NADH]/[NAD^+]}{[NADPH]/[NADP^+]}$) for the different growth regimes when demanding the respective (Sub)MDF value (see also Methods). As can be seen in Fig. 4, the well-known in vivo trend ([NADH]/[NAD⁺] < [NADPH]/[NADP⁺], i.e. $\frac{[NADH]/[NAD^+]}{[NADPH]/[NADP^+]} < 1$) is indeed predicted with this approach for all scenarios with wild-type specificity. In fact, in the great majority of cases, the ratio of reduced to oxidized cofactor is more than 1000 times higher for the NADP(H) pool compared to NAD(H), which agrees well with measurements (~1500 measured for *E. coli* in Bennett et al.[2]). The largest possible ratio (~0.85) was computed for the highest growth rate under anaerobic conditions (for SubMDF). Here it should be noted again that the (Sub)MDF values for maximal growth rates are generally the lowest (Fig. 3) implying that the thermodynamic constraints on the redox cofactor pools may become less restrictive because other thermodynamic bottlenecks (involved in the growth-rate optimal pathways) are dominating.

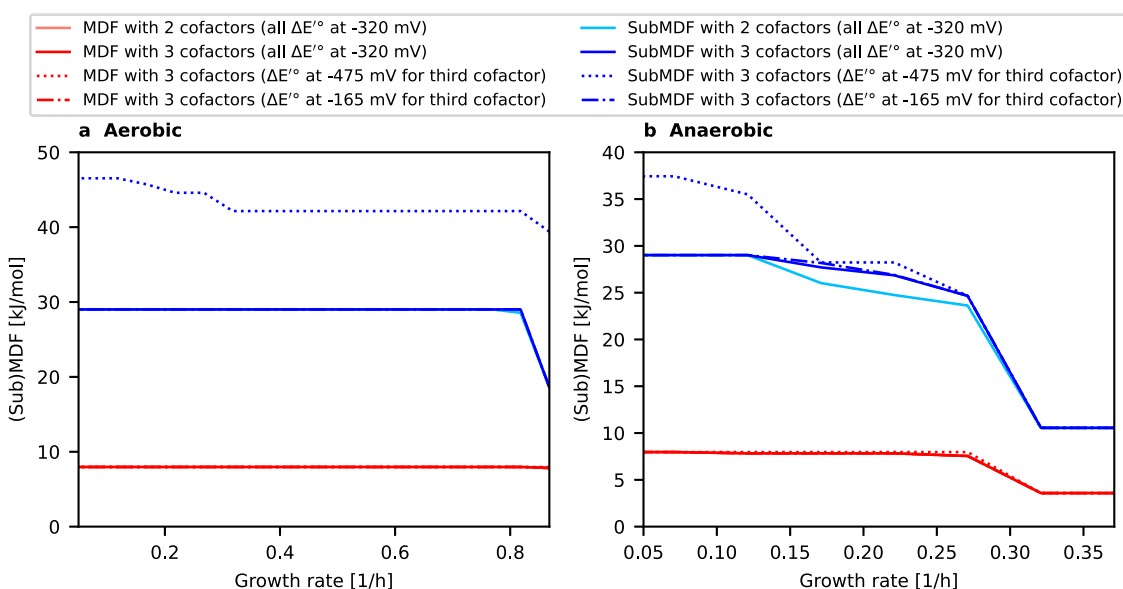

**Fig. 5 | Comparison of a model with two and of a model with three redox cofactor pools with regard to their reachable (Sub)MDF values (fully flexible specificity scenario).** For the third redox cofactor, three different scenarios regarding its redox potential were considered (see text). **a** Aerobic conditions. **b** Anaerobic conditions. Note that all red lines (associated with MDFs) coincide at least partially or even completely with the dark red line and, likewise, some blue lines (associated with SubMDFs) with the dark blue line. Source data are provided as a Source Data file.

## Achievable MDF with a third cofactor pool

Next we investigated whether the metabolic network of *E. coli* equipped with three (instead of two) redox cofactor pools could further enhance the thermodynamic driving forces in the network. Accordingly, with a fully flexible specificity for the redox cofactors, we compared the (Sub)MDF values achievable in the two-cofactor model vs. an expanded model with three possible redox cofactor pools (see Methods). An important aspect when introducing the third hypothetical cofactor is the choice of its standard redox potential ($\triangle E'^{\circ}$). For the two redox couples NADH/NAD$^+$ and NADPH/NADP$^+$, the redox potential is identical at −320 mV (corresponding to a standard formation energy ($\triangle_f G'^{\circ}$) difference of 61.75 kJ/mol between reduced and oxidized cofactor variant). We considered three different scenarios for the redox potential of the third cofactor: (1) identical to NAD(P)H/NAD(P)$^+$ ($\triangle E'^{\circ}$ = −320mV), which mimics the situation where the additional redox cofactor has very similar properties as NAD(P)(H); (2) $\triangle E'^{\circ}$ = −165mV (redox potential increased by 155 mV relative to NAD(P)H/NAD(P)$^+$; and (3) $\triangle E'^{\circ}$ = −475mV (redox potential decreased by 155 mV relative to NAD(P)H/NAD(P)$^+$; see also Methods).

The results are shown in Fig. 5. It can clearly be seen that the additional cofactor pool provides only very limited advantages if the same redox potential is used. In particular, not a single MDF advantage over the two-cofactor model can be found and only minor benefits can be observed for SubMDF for a few growth rates under anaerobic and for a single growth rate under aerobic conditions. A similar result is achieved if the redox potential of the third cofactor is increased by 155 mV to −165 mV, relative to NAD(H) and NADP(H): only marginal SubMDF advantages can be observed for aerobic conditions compared to the case with the two cofactors NAD(H) and NADP(H).

In contrast, if the redox potential of the third redox couple is decreased by 155 mV to −475 mV, the SubMDF increases significantly compared to the two-cofactor model by roughly 15 kJ/mol (aerobically) and 7 kJ/mol (anaerobically). In addition, minor MDF advantages occur at some growth rates for both conditions.

## Robustness of the results

Although our described investigations required only few inputs, namely network stoichiometry, standard Gibbs free energies and feasible metabolite concentration ranges, the results of our computations may be affected by uncertainties, especially in the used concentration ranges and the $\triangle_r G'^{\circ}$ values retrieved from the eQuilibrator API[11]. Some workarounds had to be used for $\triangle_r G'^{\circ}$ values that were not available (see Methods). Furthermore, thermodynamic driving forces cannot take into account mechanisms that can mitigate thermodynamic bottlenecks, such as the binding of a great amount of NAD(H) or NADP(H) to enzymes[12] or metabolite channeling[13]. To analyze the uncertainty of our findings with respect to altered $\triangle_r G'^{\circ}$ values, we performed most computations again with $\triangle_r G'^{\circ}$ values sampled around their initial values in a thermodynamically consistent manner (see Supplementary Note 2). From the results of 100 samples, we could conclude that the major findings of this work still hold. In particular, the difference between the maximal driving forces of the wild type and the flexible specificity still remain low in the majority of cases while random specificities have, on average, significantly reduced (Sub)MDF values (Supplementary Fig. 1).

Another uncertainty of thermodynamic-based analyses in stoichiometric models as used herein are the broad ranges of metabolite concentrations typically allowed in the calculations. We therefore also tested the effects of tighter metabolites concentration ranges using measurements reported in Bennett et al.[2] Again, as shown in Supplementary Fig. 2 and discussed in Supplementary Note 3, key results also hold if we use these significantly tighter constraints on concentration ranges. This, together with the results from the sampled $\triangle_r G'^{\circ}$ values, demonstrates the robustness of our results and provides confidence about their significance.

## Results with acetate as substrate

So far, glucose was used as substrate in all calculations. In order to check if the main findings also hold true with a different carbon source, we repeated several analyses with acetate as sole substrate. Acetate enters the central metabolism at a rather distant node relative to glucose and requires for growth gluconeogenesis instead of glycolysis. For these calculations, glucose uptake was blocked and the maximum acetate uptake rate was set to 10 $\frac{mmol}{gDW \cdot h}$, resulting in a maximal growth rate of ca. 0.21 h$^{-1}$. We considered only aerobic growth, since *E. coli* cannot grow on acetate under anaerobic conditions. All other settings remained the same as with glucose.

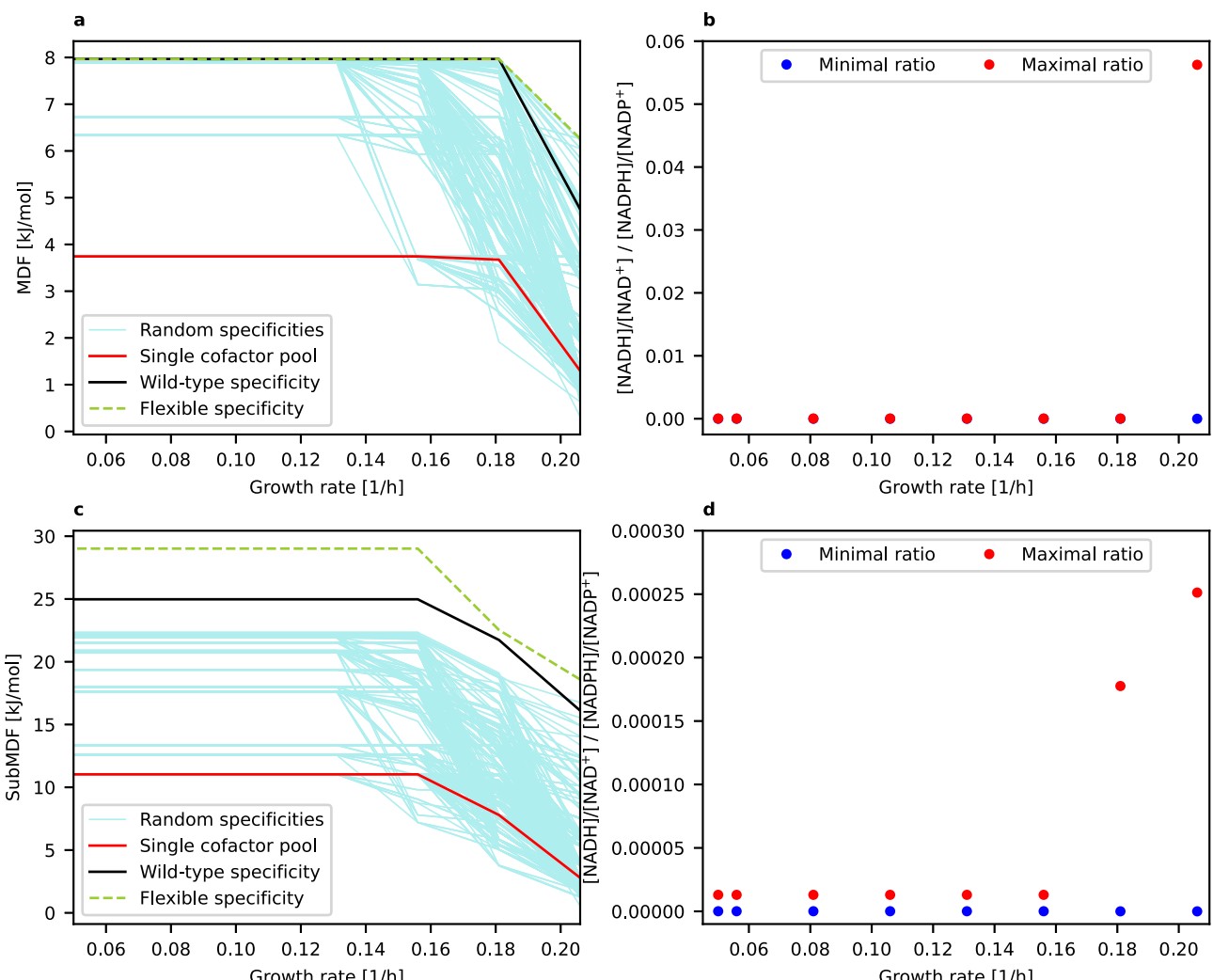

**Fig. 6 | Maximal thermodynamic driving forces (MDF and SubMDF) achievable with different NAD(P)(H) specificities and feasible ranges of the $\frac{[NADH]/[NAD^+]}{[NADPH]/[NADP^+]}$ ratio to reach the respective (Sub)MDF value of wild-type specificity (all with acetate as carbon source under aerobic conditions). a** MDF results. **b** Ratio results with MDF as target. **c** SubMDF results. **d** Ratio results with SubMDF as target. Note that if a blue dot (minimal ratio) is not seen then it largely overlaps with the red dot (maximal ratio) and is thus very close to this value. Source data are provided as a Source Data file.

The resulting (Sub)MDF values for the different specificity scenarios (Fig. 6a, c) as well as the feasible ranges of the respective $\frac{[NADH]/[NAD^+]}{[NADPH]/[NADP^+]}$ ratios (Fig. 6b, d) lead to the same main conclusions as with glucose as substrate. In particular, for almost all growth rates analyzed, the wild-type specificity reaches higher (Sub)MDF values than all random specificities. The only exception occurs at the maximal growth rate where just 25 (SubMDF) or just one (MDF) of the 1000 random specificities enable higher thermodynamic potentials. Regarding the feasible ranges of the $\frac{[NADH]/[NAD^+]}{[NADPH]/[NADP^+]}$ ratio, they again stay always far below 1, as expected.

In addition, we found that the wild-type specificity is always at most a single cofactor swap away from reaching the respective (Sub)MDF value of the flexible specificity. In most cases, the affected reaction is the reverse (gluconeogenic) direction of the glyceraldehyde-3-phosphate dehydrogenase (GAPDH) reaction

$$1,3-\text{bisphospho}-\text{D}-\text{glycerate} + H^+ + NADH$$
$$\rightarrow \text{glyceraldehyde}-3-\text{phosphate} + NAD^+ + Pi.$$

The switch towards NADP(H) could significantly increase the thermodynamic driving force of this reaction, because the ratio of NADPH (a substrate of the reaction) and NADP$^+$ (product) is significantly higher than the ratio of NADH and NAD$^+$ (see also Fig. 6b, d). This, in turn, can enhance the (Sub)MDF of the entire flux distribution. This finding is also consistent with the fact that the metabolism of many other organisms operating the GAPDH reaction in the reducing direction (e.g. in the Calvin cycle of photosynthetic organisms) prefers NADPH as cofactor. It would therefore be interesting to investigate whether a cofactor swap for this reaction in *E. coli* can enhance growth on acetate.

We also reconsidered the reactions of the PDH and the iso-citrate dehydrogenase regarding the effects of altered NAD(P)(H) specificity. Regarding PDH, a SubMDF advantage can be found in one growth rate, while a cofactor swap of the isocitrate dehydrogenase would not provide any (Sub)MDF advantage with acetate as substrate, in fact, it would reduce the (Sub)MDF for some growth rates. This example indicates again that understanding the preferred redox cofactor specificity in the wild type may, at least partially, require the consideration of all relevant growth conditions.

## Discussion

In the present study, we developed and employed the TCOSA framework to study key aspects of redox cofactor redundancy in metabolic networks. TCOSA uses an unbiased approach, solely based on a stoichiometric model, thermodynamic constraints (Gibbs free energies) and metabolite concentration ranges. It attempts to relate NAD(P)(H) reaction specificity distributions to network-wide thermodynamic driving forces. Our main findings, obtained in a genome-scale metabolic model of *E. coli* for varying growth rates under aerobic as well as anaerobic conditions, can be summarized as follows: (1) A redundant set of two redox cofactor pools is clearly beneficial for maximizing the thermodynamic driving force in the network, while a third redox cofactor would bring a significant advantage only if it has a different redox potential than NAD(P)H. (2) With TCOSA we can show that evolved NAD(P)(H) coenzyme specificities of metabolic reactions in *E. coli* are likely the consequence of a global (network-wide) maximization of thermodynamic driving forces since the maximal thermodynamic potential (quantified as MDF) achievable with the wild-type specificities lies often close to or even equals the theoretical optimum. (3) Trends in optimal redox-cofactor concentration ratios can be predicted by our approach reflecting the in vivo situation. (4) The obtained results are robust against variations of model parameters (standard Gibbs free energies and metabolite concentration ranges).

The most striking result of our analysis is that the wild-type distribution of NAD(P)(H) specificities of redox reactions in *E. coli* enables higher overall thermodynamic driving forces than almost all generated random specificities (Fig. 3). In fact, in the majority of the cases, the wild-type specificities even reach the maximum (Sub)MDF values achievable with a (hypothetical) fully flexible specificity. This result is remarkable and indicates that redox cofactor specificities are likely an evolutionary result of an optimization principle that maximizes the global thermodynamic driving force as represented by (Sub)MDF. The evolutionary advantage of higher (Sub)MDF values lies in the fact that low driving forces possibly limiting the flux of certain essential reactions become less likely. The relevance of the thermodynamic driving force for the reaction flux can be seen if the typical kinetic function of a reaction flux is decomposed[14]. Moreover, reactions operating closer to thermodynamic equilibrium increase the demand of enzymes to reach required flux levels[3]. Since high MDF values indicate potentially high thermodynamic driving forces for all reactions, high enzyme demands due to thermodynamic limitations become less likely as well. In addition to thermodynamic effects, the NAD(P)(H) distribution can also affect the possible stoichiometric balances of NAD(H) and NADP(H) producing and consuming reactions. A change of this distribution may require the activation of new pathways with potentially lower or higher thermodynamic driving forces. These effects are directly taken into account in our TCOSA approach.

The fact that the wild-type specificity does not always reach the theoretically maximal (Sub)MDF does not necessarily contradict our main conclusion. First of all, it should be emphasized again that under the flexible specificity scenario, where the maximal (Sub)MDF values are reached, the specificities can change under different growth rates and conditions, which is obviously not possible in a fixed (wild-type) distribution. Therefore, since we found that the flexible specificities change under varying conditions, any fixed specificity must perform worse than the fully flexible scenario. Furthermore, other constraints that cannot be considered by our framework may affect the optimal NAD(P)(H) specificity and the achievable MDF. For example, certain metabolite levels required for MDF-optimal flux distributions might be very high and thus possibly toxic confining the feasible ranges of metabolic concentrations to a much narrower region than assumed herein. Moreover, feasible (ranges of) metabolite concentrations are to a large extent determined by kinetic properties of the reaction mechanisms and enzymes including allosteric (and genetic) regulation of enzymes. Due to those factors, it appears likely that the computed

MDF of the wild-type specificity is not reached in vivo. Nevertheless, analyzing MDF-optimal solutions is still meaningful in the sense that the computed MDF indicates the thermodynamic flexibility of the system: a high MDF value indicates that the metabolite concentrations can be chosen from a broader range to keep the flux distribution at higher driving forces or at least thermodynamically feasible ($\triangle_r G' < 0$ for all active reactions)−even if the theoretically feasible MDF itself is not reached. Indeed, despite the mentioned potential limitations, our MDF-based modeling framework turned out to be very useful to address various aspects relevant for cofactor specificity.

Another important finding was that, in order to reach the theoretically optimal (Sub)MDF of the flexible specificity, only relatively few reactions of the wild-type specificity would need to swap their cofactor usage (Table 1). Interestingly, this analysis suggested, for example, a cofactor swap for the isocitrate dehydrogenase from NADP(H) to NAD(H) and thus to a cofactor specificity that is indeed known for this enzyme from other organisms. The example of growth on substrate acetate (instead of glucose), where NADP(H) dependency seems to be advantageous[9], demonstrates that other factors not covered herein may contribute to the manifestation of the cofactor specificity found in *E. coli*.

For the majority of cases, our single-swap analysis showed either a negative or no effect on the respective MDF. The fact that several single swaps had no effect implies that there are many cofactor specificity distributions which are as good as the wild-type specificity. Hence, while the wild-type specificity can, to a large extent, be explained through thermodynamic favorability, it cannot be precisely predicted since equally good alternative specificities are possible. However, the overall likelihood of randomly finding a specificity enabling the high (Sub)MDF values as in the wild type is very low as shown by the random specificity sampling. Again, one may also speculate that many of these alternative specificities may have certain disadvantages under particular conditions that are not covered by our approach.

We used our modeling approach to study several further aspects related to redundant redox cofactor pools. First, we showed that qualitative trends of redox cofactor ratios can be predicted, i.e. that the NAD(H) pool is typically much more oxidized than the NADP(H) pool. We also employed our method to determine the optimal number of cofactor couples. The results obtained with this analysis clearly demonstrated that using a single cofactor pool would largely decrease the thermodynamic driving forces in the network. This confirms again the postulated advantage of having two cofactor pools[3], since this allows the cell to have one largely oxidized and one largely reduced cofactor pool and the specificity of each reaction can then be chosen such that the network-wide driving force is optimized. On the other hand, analyzing a three-cofactor model, where a hypothetical third cofactor couple aside of NAD(H) and NADP(H) can be used, revealed that significant benefits for the (Sub)MDF are only possible if the redox couple of the third cofactor has a redox potential that is significantly different from NAD(P)H/NAD(P)[+]. This may explain the existence of other redox cofactors in *E. coli* such as thioredoxins. However, the latter is involved in much less metabolic reactions than NAD(P)(H) and it can be speculated that the relevance of electron carriers like thioredoxin lies more in enabling specialized electron transfer reactions (here: reduction of disulfide bonds of proteins). For organisms with complicated metabolic lifestyles, a third widely used redox cofactor may pay off even more. For example, carbon-fixing anaerobic acetogens (representatives of chemolithoautotrophic microorganisms) use, in addition to NAD(H) and NADP(H), ferredoxin (with a more negative redox potential) as a third major electron carrier in many redox reactions of the central metabolism giving them an extra degree of freedom to balance their complex redox metabolism[15].

Some aspects of our work are related to the study of Goldford et al.[3], where the authors sought to find explanations for the presence

of cofactor redundancy (with focus on NAD(H) and NADP(H)). For some of the performed analyses, this study also used a stoichiometric model of *E. coli* with thermodynamic constraints. The key difference to our TCOSA framework is that they analyzed merely the thermodynamic feasibility of metabolic flux distributions in this network (partially with fixed NAD(P)H/NAD(P)$^+$ ratios) while we maximized for a network-wide thermodynamic driving force (MDF) allowing a broad interval for the concentrations of all metabolites, including the cofactors. Nevertheless, there are some overlapping results, which are largely consistent. The authors in Goldford et al.[3] used, in addition to the stoichiometric model, a minimal (fictive) enzyme-cost model with sampled kinetic parameters to show that redundant redox cofactor pools enable higher thermodynamic driving forces eventually lowering the enzyme costs. This finding is consistent with our conclusion, here solely derived from network stoichiometry and basic thermodynamic constraints alone, that two redox couples may increase the maximal achievable thermodynamic driving forces in the network (which, in turn, implies reduced enzyme costs). Most importantly, the key difference and major advancement of our approach lies in the fact that we can determine optimal NAD(P)(H) specificities maximizing the thermodynamic potential of a given metabolic reaction network and compare these with the wild-type specificities in *E. coli* to learn how thermodynamic constraints have shaped the redox cofactor specificities in the metabolic network of *E. coli*. As a more stringent result compared to the flux-based calculations reported in Goldford et al.[3], our stoichiometric thermodynamic-based approach indicates that a significant number of individual cofactor swaps, even if they are not lethal, can still have an effect on the achievable thermodynamic driving force in the *E. coli* metabolic network and that there is a significant number of such swaps that would be unfavorable. This suggests again that the redox cofactor specificity of metabolic reactions in *E. coli* has evolved to enable high thermodynamic potentials.

As part of our TCOSA framework, a number of tools and algorithms have been developed, which are available from a public code repository and can be used to reproduce the results of this work or to perform similar analyses with other metabolic networks. Moreover, we envision that these developments can also be useful for several other applications. One such application, related to the metabolic engineering of cell factories for bio-based production processes, could be to find optimal NAD(P)(H) specificities that maximize the thermodynamic driving force towards a product of interest. Other theoretical studies have already addressed the computation of optimal cofactor swaps in order to maximize product yields[16–21]. Our approach can extend these techniques with thermodynamic considerations such that targeted changes in cofactor specificities do not only increase product yields but also enable higher driving forces along the pathway to the product. A targeted change of redox cofactor specificities has come into reach by several experimental techniques, including direct protein structure changes[22], and adaptive evolution methods[23,24].

A thermodynamic optimization strategy related to the TCOSA approach was recently proposed for the design of microbial communities with improved thermodynamic driving forces for the production of chemicals[25]. The basic idea of this approach resembles the existence of two redox cofactor pools, namely to have different pools of one metabolite (or of one functional class of metabolites) with different *in vivo* thermodynamic potentials caused by different metabolite abundances. For example, if a product pathway involves a metabolite in one reaction as substrate and in another as (by)product, it would be desirable to have this metabolite in the first reaction in high and in the second reaction in low concentrations. One way to achieve this is compartmentalization. While it appears difficult to build such compartments with desired properties in a cell, specifically designed microbial co-cultures could be used through which different metabolite concentrations in the involved strains or species would become

possible to maximize the overall thermodynamic driving force along the product pathway[25]. In this line, our work may also extend investigations on the evolution of compartmentalization[26,27] with respect to thermodynamic aspects. Another potential application of methods and tools developed herein is to study thermodynamic aspects of other cofactor redundancies in the metabolism, e.g. of ATP and GTP[3]. Finally, our introduced definition of SubMDF, opposed to MDF, can be useful for other thermodynamics-based studies of metabolic pathways (or subnetworks), as it ensures global thermodynamic feasibility of entire flux distributions while maximizing thermodynamic driving forces only for the actual parts of interest.

## Methods

### Metabolic model and TCOSA-related reconfigurations

As basis for all calculations in this study, we used *i*ML1515, the latest genome-scale metabolic model of *E. coli*[6], which uses BiGG IDs[28]. We extended and modified this model as described in the following to prepare it for TCOSA-related analyses (see also Fig. 1A):

- Thermodynamic constraints: For thermodynamic calculations, we used standard Gibbs free energies of reactions ($\triangle_r G'^\circ$) calculated using a custom script that accesses the eQuilibrator Python API[11]. As settings for this calculation, we used a pMg of 2.5, an ionic strength of 250 mM, a cytosolic pH of 7.5 and, for reactions crossing membranes (multi-compartmental reactions), a standard membrane potential difference of 150 mV and a ΔpH of +0.5 between cytoplasma and periplasma. For the growth pseudo-reaction and all (331) exchange pseudo-reactions with the environment, no $\triangle_r G'^\circ$ was set (i.e., they are unconstrained). Out of the remaining 2712−332 = 2380 reactions in the iML1515 model, a $\triangle_r G'^\circ$ could be calculated for 1683 reactions. For the remaining 697 reactions where a $\triangle_r G'^\circ$ could not be found or determined, a low $\triangle_r G'^\circ$ of −100 kJ/mol was set to impose at least some minimum constraints on the associated metabolite concentrations. The same procedure was used for 611 multi-compartmental reactions whose role was solely classified as transporter (no conversion takes place), to consider pure transport of metabolites or nutrients as thermodynamically (almost) unlimited. This does not hold for reactions such as of the ATP synthase or NADH dehydrogenase, where proton translocation is coupled with ATP synthesis or NADH oxidation, respectively. Due to their importance for this study, a special treatment was done for NAD(P)(H)-dependent reactions. For 161 of the 238 original NAD(P)(H)-dependent reactions in *i*ML1515 (68%), a $\triangle_r G'^\circ$ could be calculated. For the other 77 cases, the following $\triangle_r G'^\circ$ values were used to better reflect realistic values: from the 161 reactions with known $\triangle_r G'^\circ$ we determined median $\triangle_r G'^\circ$ values of all reactions where NAD$^+$ ( + 15.79 kJ/mol), NADH (−15.79 kJ/mol), NADP$^+$ ( + 15.33 kJ/mol), or NADPH (−15.33 kJ/mol) is a substrate, respectively. Depending on cofactor usage, we then used these values for the NAD(P)(H)-dependent reactions with unknown $\triangle_r G'^\circ$. For an analysis of the effect of uncertainties in the $\triangle_r G'^\circ$ values see also Supplementary Note 2.
- For the metabolites, a standard concentration range from $10^{-6}$ M to 0.02 M was used as in Hädicke et al.[5] Concentrations of protons (H$^+$) and water were fixed at 1 M due to their inclusion in the $\triangle_r G'^\circ$ calculation. Calculations with tighter constraints for the metabolite concentrations based on in vivo data[2] are discussed in Supplementary Note 3.
- In principle, with inclusion of thermodynamic constraints, one could consider all reactions reversible and then let the concentration vector and resulting Gibbs free energies decide which direction is taken in a given solution. However, especially in cases where the standard Gibbs free energies are not known, this may lead to unrealistic flux directions and we therefore kept the original reversibilities.

- The ermodynamic calculations as performed herein can often be simplified if all reactions can operate in only one direction. Therefore, we split all reversible reactions into two irreversible (forward and backward) reactions (the old ID is changed to "ID_FWD" and "ID_REV", respectively). The $\triangle_r G'^\circ$ values are negated in the reverse direction except for the mentioned reactions whose (unknown) $\triangle_r G'^\circ$ value was set to −100 kJ/mol; in this case −100 kJ/mol was also used for the reverse direction to not favor any of the two directions.
- We found that growth is thermodynamically infeasible in the $i$ML1515 model with the assigned $\triangle_r G'^\circ$ values and metabolite concentration ranges. As detailed in the Supplementary Note 1, we therefore computed the minimal number of reactions whose $\triangle_r G'^\circ$ values have to be relaxed in order to make growth thermodynamically feasible. In total, the thermodynamic constraints of 9 reactions had to be relaxed, none of which is involved in the central carbon metabolism.
- Next, as essential part of the TCOSA framework, for each reaction that uses NAD$^+$ (BiGG ID nad_c) $and$ NADH (nadh_c), the reaction ID was changed to "ID_ORIGINAL_NAD_TCOSA" and a duplicate reaction named "ID_VARIANT_NADP_TCOSA" was introduced where NAD$^+$ was replaced with NADP$^+$ (nadp_c) and NADH with NADPH (nadph_c). Likewise, for each reaction that involves NADP$^+$ and NADPH, the reaction's ID was changed to "ID_ORIGINAL_NADP_TCOSA" and a duplicate reaction named "ID_VARIANT_NAD_TCOSA" was introduced where NADP$^+$ was replaced with NAD$^+$ and NADPH with NADH. As one exception, in reactions where both NAD(H) and NADP(H) occur simultaneously (such as in the NAD kinase, in the transhydrogenase, or in the growth pseudo-reaction), no duplication occurs. The described reconfiguration with two cofactor variants is in large parts analogous to the model modifications used by the OptSwap method[18], but analyses with OptSwap focused solely on selected oxidoreductases and did not consider thermodynamic driving forces.

We call the resulting model $i$ML1515_TCOSA. The original $i$ML1515 model contains 128 reactions involving NADH and NAD$^+$ (thereof 34 reversible), 110 reactions involving NADPH and NADP$^+$ (thereof 27 reversible) as well as 6 irreversible reactions involving all four cofactor variants (transhydrogenase, growth reaction etc.). In total, with all split reversible reactions and the introduced cofactor duplicate reactions, $i$ML1515_TCOSA consists of 3982 reactions and 1881 metabolites, thereof 299 reactions involving NAD$^+$ and NADH, another 299 reactions involving NADP$^+$ and NADPH, and (as before) 6 reactions involving NAD(H) and NADP(H) simultaneously.

We also created an $i$ML1515_TCOSA variant, called $i$ML1515_3TCOSA, where NAD(P)H-containing reactions where not only duplicated but even triplicated. In this model, the third reaction copy is called "ID_VARIANT_NADZ", where NAD(P)$^+$ was replaced by a new hypothetical cofactor NADZ$^+$ (nadz_c), and NAD(P)H by the new (reduced) cofactor variant NADZH (nadzh_c). For the cases where the redox potential of the third cofactor was decreased (increased) by 155 mV relative to the redox couples NAD(P)$^+$/NAD(P)H, the $\triangle_r G'^\circ$ of all reactions with the third cofactor variant was increased (decreased) by a corresponding value of 30 kJ/mol multiplied with the stoichiometry of the oxidized cofactor variant.

In all simulations, potential release of (e.g. fermentation) products via exchange reactions was allowed as in the original $i$ML1515 model and the minimum ATP maintenance demand (ATPM reaction) was kept at 6.86 $\frac{mmol}{gDW \cdot h}$. For growth on glucose we assumed a maximum glucose uptake rate of 10 $\frac{mmol}{gDW \cdot h}$ for aerobic conditions and 20 $\frac{mmol}{gDW \cdot h}$ for anaerobic conditions. For growth on acetate (under aerobic conditions) a maximum acetate uptake rate of 10 $\frac{mmol}{gDW \cdot h}$ was set.

## MDF

Most computational analyses performed in this work are based on the mixed-integer linear program (MILP) OptMDFpathway[5], which can be used to find, in a given metabolic reaction network, the steady-state flux distribution with maximal thermodynamic driving force obeying given constraints including flux bounds, metabolite concentrations, and standard Gibbs free energies. A full description of this method can be found in Hädicke et al.[5] Briefly, a metabolic network with $m$ (internal) metabolites and $q$ reactions is represented by the $m \times q$ stoichiometric matrix $\mathbf{N}$. The rate vector $\mathbf{r}$ contains the $q$ reaction fluxes and demanding steady state for the metabolite concentrations implies the mass balance constraint

$$\mathbf{N}\mathbf{r} = \mathbf{0}. \tag{1}$$

The flux of a reaction $i$ is constrained by flux bounds

$$\alpha_i \leq r_i \leq \beta_i. \tag{2}$$

More complex linear constraints for the fluxes (e.g., enzyme constraints[29]) can be included via a suitable pair of a matrix $\mathbf{D}$ and a vector $\mathbf{d}$:

$$\mathbf{D}\mathbf{r} \leq \mathbf{d}. \tag{3}$$

Thermodynamic constraints can be incorporated via the reaction-centric driving forces ($f_i$) defined as the negated Gibbs free energy ($\triangle_r G_i'$) of reaction $i$:

$$f_i = -\triangle_r G_i' = -\triangle_r G'^\circ - RT \cdot (\hat{\mathbf{N}}_{\cdot,i})^T \cdot \mathbf{x}. \tag{4}$$

$R$ is the gas constant, $T$ the temperature (here 298.15 K), $\mathbf{x}$ a vector containing the logarithmized metabolite concentrations and $(\hat{\mathbf{N}}_{\cdot,i})^T$ is the transposed $i$-th column (reaction) of the extended stoichiometric matrix $\hat{\mathbf{N}}$, which contains, in addition to $\mathbf{N}$, also the external metabolites in its rows for which the steady-state balance (Eq. 1) does not apply. $f_i$ indicates directly whether operation of reaction $i$ in forward direction is thermodynamically feasible ($f_i > 0$) or not ($f_i \leq 0$) and the value of $f_i$ is a measure for the driving force of the reaction. Note that $f_i$ is limited by the (allowed) ranges of metabolite concentrations specified by vectors of lower ($\mathbf{c_{min}}$) and upper ($\mathbf{c_{max}}$) bounds for the concentrations, which in turn also constrain the logarithmized concentrations in $\mathbf{x}$:

$$\ln(\mathbf{c}_{min}) \leq \mathbf{x} \leq \ln(\mathbf{c}_{max}) \tag{5}$$

The measure of thermodynamic driving force (and feasibility) of a single reaction ($f_i$) can be extended to a pathway by using the concept of max−min driving force (MDF)[4], which is defined as the maximal value $B$ such that $f_i \geq B$ for all reactions of a given pathway (Fig. 2). While the original MDF definition is useful to quantify the maximal driving force of a given pathway, the OptMDFpathway approach[5] generalizes this concept: for an entire metabolic network and a demanded phenotypic behavior (e.g., a given growth rate) it searches for a steady-state flux distribution and suitable metabolite concentrations that maximize the MDF for the predefined phenotypic behavior (Fig. 2). Herein we always refer to this generalized MDF (associated with a given network and demanded behavior or condition), which can be determined as follows. For each reaction $i$, a binary variable $z_i \in \{0,1\}$ is defined, which must become 1 if the reaction carries a positive flux:

$$r_i \leq z_i \cdot \beta_i. \tag{6}$$

This formulation requires all reactions to be irreversible ($r_i \geq 0$), which is the reason why all reversible fluxes are split into two

irreversible ones in a preprocessing step (see above). For the MDF $B$ of a given flux distribution we have to demand that the driving force of every *active* reaction is equal or higher than $B$:

$$B \leq f_i + M \cdot (1 - z_i), \tag{7}$$

where $M$ is a large number lying above the maximal feasible driving force of any reaction.

As explained above, in the reconfigured TCOSA model, a reaction is split if it is reversible or/and it is duplicated with swapped cofactor specificity if it uses NAD(P)H (see also Fig. 1). Additional constraints are added in order to prevent the promiscuous use of both NAD(H) *and* NADP(H) reaction variants (only one can be used at a time): for every reaction $k$ of the original *i*ML1515 model we built an index set $A_k$ containing the indices of all reaction variants in the TCOSA model derived from this original reaction (in the case of a reversible NAD(P)(H)-using reaction, there can be up to 4 such indices). Then, for each $A_k$ with two or more indices, the following constraint is added:

$$\sum_{i \in A_k} z_i \leq 1. \tag{8}$$

Finally, the MDF value (and its associated flux distribution and metabolite concentrations) for the given network and constraints can then be found by a linear objective function maximizing the MDF $B$:

$$\begin{aligned} & \underset{\boldsymbol{x}, \boldsymbol{r}, \boldsymbol{z}}{Maximize} \quad B \\ & s.t.\, eqs.\, (1)-(8) \end{aligned} \tag{9}$$

We note that the MILP optimization problem for the computation of MDF (Eqs. (1)–(9)) overlaps to some extent with the MILP formulations of thermodynamics-based (metabolic) flux analysis (T(M)FA) and thermodynamics-based flux balance analysis (TFBA), which seek to analyze thermodynamically consistent flux distributions[30]. The key difference is that TFBA and T(M)FA explicitly demand that the Gibbs free energy of all reactions is negative, hence, that the reactions obey the second law of thermodynamics. Within these constraints, TFBA is used to optimize certain fluxes (e.g. growth rate as usual for FBA), metabolite concentrations or particular Gibbs free energies. In our MDF formulation, the objective function (9) maximizes the MDF $B$, i.e. it maximizes the minimal driving force of all active reactions. The Gibbs free energies are hidden behind the driving forces $f_i$ and the MDF $B$ (the latter capturing the smallest of all relevant driving forces). In most calculations, we did not explicitly demand that $B>0$ (which would imply $\triangle_r G_i' < 0$ for all reactions) but one can easily check feasibility of the found solutions by verifying that $B>0$. If $B \leq 0$ then the absolute value of $B$ indicates the distance of the solution from thermodynamic feasibility.

## SubMDF

We introduce SubMDF, which represents the MDF achievable within a certain subset (instead of all) of the active reactions. In our application, the relevant subset comprises the NAD(P)(H)-dependent reactions. By definition, the SubMDF is at least as high as the MDF. The considered subnetwork is represented by an index set $S$ of its reactions. When maximizing the MDF in a certain part of the network, we still need to ensure that all *active* reactions (also in the rest of the network) remain feasible. Therefore, in addition to the introduced constraints for MDF, we first demand

$$B \geq B_{\min} \tag{10}$$

with $B_{\min}>0$ to ensure thermodynamic feasibility of the found solution in the entire network. Herein, we used $B_{\min} = 0.1$ kJ/mol. We then introduce $B_{sub}$ denoting the MDF in the subnetwork, which must obey

$$B_{\text{sub}} \leq f_j + M \cdot \left(1 - z_j\right), \forall j \in S. \tag{11}$$

By maximizing for $B_{\text{sub}}$ we obtain SubMDF:

$$\begin{aligned} & \underset{\boldsymbol{x}, \boldsymbol{r}, \boldsymbol{z}, B}{Maximize} \quad B_{\text{sub}} \\ & s.t.\, eqs.\, (1)-(8),(10),(11) \end{aligned} \tag{12}$$

## Analysis of metabolite concentration ratios

For a relative comparison of the feasible [NADH]/[NAD$^+$] concentration ratio (denoted by $P_{NADH}$) with the [NADPH]/[NADP$^+$] ratio ($P_{NADPH}$), we performed variability analyses of the ratio of $P_{NADH}$ and $P_{NADPH}$ defined as $\frac{P_{NADH}}{P_{NADPH}} = \frac{[NADH]/[NAD^+]}{[NADPH]/[NADP^+]}$. The logarithmized version of this ratio can be written as

$$\rho = \ln\left(\frac{[NADH]/[NAD^+]}{[NADPH]/[NADP^+]}\right) = x_{\text{NADH}} + x_{\text{NADP}^+} - x_{\text{NADPH}} - x_{\text{NAD}^+}. \tag{13}$$

In our analyses, the MDF value is fixed to the previously found maximal MDF ($B_{\max}$) or SubMDF ($B_{sub,\max}$) value by setting:

$$B = B_{\max} \tag{14}$$

or

$$B_{sub} = B_{sub,\max}. \tag{15}$$

We can then determine the maximum and minimum value for $\rho$ via the two optimization problems

$$\begin{aligned} & \underset{\boldsymbol{x}, \boldsymbol{r}, \boldsymbol{z}}{Maximize} \quad \rho \\ & s.t.\, eqs.\, (1)-(8),\, \text{and}\, (14)\, \text{or}\, (15) \end{aligned}$$

and

$$\begin{aligned} & \underset{\boldsymbol{x}, \boldsymbol{r}, \boldsymbol{z}}{Minimize} \quad \rho \\ & s.t.\, eqs.\, (1)-(8),\, \text{and}\, (14)\, \text{or}\, (15) \end{aligned}$$

and then take $e^{\rho,\min}$ and $e^{\rho,\max}$ to obtain the minimal and maximal value of $\frac{P_{NADH}}{P_{NADPH}}$.

## Minimal cofactor swap analysis

To determine the minimal number of reactions that have to swap their (wild-type) NAD(P)(H) specificity in order to reach the calculated MDF or SubMDF values in the flexible specificity scenario (where arbitrary cofactor swapping is allowed), we proceed as follows: First, we collect the indices of all reactions which have the complementary NAD(H) and NADP(H) specificity compared to the wild type in an index set $C$. For all these reactions, a new set of binary variables $z_{i,swap} \in \{0,1\}, i \in C$, is introduced allowing the reaction in the wild type to change its specificity:

$$r_{i,swap} \leq z_{i,\text{swap}} \cdot \beta_i \forall i \in C. \tag{16}$$

Here, $r_{i,\text{swap}}$ is the rate of the reaction $i$ with alternative specificity, which is enabled in the TCOSA model if $z_{i,\text{swap}} = 1$. Now, to find the minimum number of cofactor swaps at previously calculated optimal (Sub)MDF values, we formulate the MILP optimization problem

$$\underset{\boldsymbol{x}, \boldsymbol{r}, \boldsymbol{z}, \boldsymbol{z_{swap}}}{Minimize} \sum z_{i,\text{swap}}$$

*s.t. eqs.* (1)–(8), (16), and (14) or (15).

## Implementation
All calculations have been performed via Python scripts under a custom Anaconda environment. For the handling of metabolic models (via SBML) and for basic flux balance analyses, COBRApy was used[31]. The Python library pulp[32] was employed for the construction of the MILPs and a recent version (12.10) of IBM CPLEX was used as MILP solver. All scripts, generated data and metabolic models (the latter in SBML format[33]) can be found, together with a documentation, in the TCOSA GitHub repository under the link https://www.github.com/klamt-lab/TCOSA and in Zenodo[34].

The calculations were performed on a computer cluster (with a total of 16 Intel Xeon Silver 4110 CPU cores and 192 GB DDR4 RAM). Sequentially running all calculations relevant for this work on this cluster took ca. 6 days.

### Reporting summary
Further information on research design is available in the Nature Portfolio Reporting Summary linked to this article.

## Data availability
The data generated in this study have been deposited in GitHub under the link https://www.github.com/klamt-lab/TCOSA and the relevant release can also be accessed via Zenodo[34]. Source data are provided with this paper.

## Code availability
The computer code and metabolic models used for the performed calculations as well as the subsequently generated data and a documentation can be found in the TCOSA GitHub repository under the link https://www.github.com/klamt-lab/TCOSA and the relevant release can also be accessed via Zenodo[34].

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

## Acknowledgements
The authors are grateful to Axel von Kamp for his assistance and discussions on MDF calculations in the context of TCOSA. This research was partially financially supported by the German Federal Ministry of Education and Research (FKZ: 031L0104B).

## Author contributions
S.K. conceived and supervised the project. P.S.B. implemented the TCOSA framework and performed the computations. P.S.B. and S.K. analyzed and discussed the results and wrote the manuscript.

## Funding

## Competing interests
The authors declare no competing interests.
