## [Peer Review File · Nature Communications]

REVIEWER COMMENTS

Reviewer #1 (Remarks to the Author):

The paper by Bekiaris and Klamt analyzes the NAD(P)H cofactor specificity of the metabolic network of *Escherichia coli*. The authors adapt an available genome-scale metabolic network for this purpose, by allowing swaps (changes) of preferred cofactors, and combine this network with available thermodynamic information for individual reactions (negative Gibbs free energy changes). Using the OptMDFpathway tool the authors predict a flux distribution maximizing the thermodynamic driving force of the network, for both the wild-type NAD(P)H specificities and different scenarios with perturbed specificities. The authors reach two very interesting main conclusions: (i) the wild-type NAD(P)H specificities give rise to a thermodynamic driving force close to the optimal, and (ii) random NAD(P)H specificities usually perform much worse than the wild-type NAD(P)H specificities.

I find the methodology followed in this study exemplary and the conclusions very interesting and thought-provoking. My comments mostly concern the presentation and the interpretation of the results.

1. I got a bit lost in the definition of the different forms of thermodynamic driving forces: max-min driving force, optimal max-min driving force, optimal max-min driving force for a network. A box or a figure intuitively explaining these concepts would help readers who are not already familiar with these concepts.

2. In the same way, the rationale of maximizing the max-min driving force is briefly explained in the Discussion, but the text would be easier to follow if the reason for optimizing this criterion was explained at an earlier stage already.

3. In comparison with MDFs, an OptMDF also contains information on fluxes and metabolite concentrations. This information is not really exploited in the text though (except for the computing the cofactor ratios). One question that I posed myself while reading the text is to which extent the predicted fluxes and concentrations for growth of wild-type *E. coli* using the model match those that have been observed experimentally? At present, only the predicted growth rate is compared with experimental data. And how does the predicted optimal thermodynamic driving force for wild-type specificities compare with the experimentally observed (global) driving force? The above comments concern the same point: how well do the predictions of the model, under the chosen objective criterion, capture the physiology of wild-type *E. coli*?

4. In the discussion of reactions having a different optimal specificity depending on the growth conditions (e.g., growth on glucose or on acetate, p10), I was wondering whether *E. coli* possesses isoenzymes with different cofactor specificities that are differentially expressed depending on the growth condition.

5. Another interesting prediction is that using three instead of two cofactor pairs does not contribute to higher thermodynamic driving forces. Is this a (logical?) consequence of the fact that a reaction has two directions and (which may come to the same thing) that cofactor recycling has a forward and a backward direction?

Reviewer #2 (Remarks to the Author):

In this work, the authors develop a thermodynamic framework to explore the theoretical basis for NADP/NAD choice and redundancy across the metabolic network of *E. coli*. This work closely mirrors a similar effort by Segre's group published last year, which the authors discuss and cite, but the current effort goes farther in terms of a global thermodynamic analysis. The authors claim that the observed split of NAD and NADP dependent reactions in a real network (that of *E. coli*) is near optimal and their in vivo concentration ratio can be predicted from network-wide optimality, which is quite a satisfying result. The paper is written very clearly and the methods are simple to understand, utilizing the established MDF thermodynamic modeling framework. The authors include a sensitivity analysis of Gibbs energies (via sampling described in the supplement) and metabolite concentrations (via in vivo measured concentrations in the Supplement). My only comments are related to the depth of the analysis, which I hope the authors can expand.

Major Comments

- It would be nice to see the single swap analysis expanded, specifically analyzing the relative impact of specific swaps to identify key reactions. Currently the finding seems to be that these swaps have little impact, quoting "our single-swap analysis showed either a negative or no effect on the respective MDF." It is not clear how negative these MDF changes are, and it's not clear that MDF offers a complete picture governing cofactor choice. For example, we never see cofactor swaps in most major enzymes like glycolysis across organisms, but the authors seem to indicate these are

possible without repercussions. Presumably the concentrations would have to change drastically to accommodate swaps as well – would these shifts cause issues? Or would cofactor regeneration (flux considerations) become an issue somehow? What is the picture when this analysis incorporates real metabolite concentrations? The PDH example is interesting in this regard. This reaction is the entryway to the TCA cycle and one would think that its production of NADH would be important to global energy production. Are there any side effects of swapping this reaction to NADPH specific, such as lower energy production capability?

- In this vein, more analysis of individual reactions with this framework would be welcome. It's expected that due to smaller flux levels, most of the 238 nad/nadp reactions would not contribute much to the MDF and thus the key reactions determining MDF would come down to a much smaller number of relevant fluxes, is that not the case?

- Related to the above issue, what is the argument for why any enzyme with the thermodynamic flexibility to pick one cofactor or another does in fact pick the observed cofactor and not the other? A general set of rules governing this choice would be a nice result of this analysis. For example, why do some reactions have dual specificity? Is it lack of importance or some state specific flexibility that is required related to the concentration ratios of the cofactors or necessity of regenerating them?

- Further theoretical results with their framework would be welcome to add depth to their analysis. For example, is there any reason that NADH is the catabolic and NADPH is the anabolic cofactor, as opposed to vice versa? Any reason for those redox potentials in particular? What if a different molecule with a different potential was chosen? Any reason that certain pathways are chosen to regenerate NADPH?

- The authors try adding a third redox cofactor and show little effect on MDF. However, there are other redox cofactors present in real networks, (e.g. thioredoxin) – any idea on why?

- It's not clear what is learned from the growth rate perturbations. The MDF is reduced at the highest growth rates, suggesting that the optimal stoichiometric pathways place additional thermodynamic pressure on this system, if I understand correctly. This is expected but I do not understand what we learn from this exercise. The differences in MDF presumably are due to pathway choices at particular growth rates, so maybe additional investigation into these pathway choices and relationship to MDF would make this analysis richer.

- Results related to aerobic vs anaerobic growth are also relatively thin, as far as contributing to the overall picture of cofactor choice. Can the authors predict the metabolite shift from aerobic to

anaerobic (e.g. using the data in McCloskey et al <https://doi.org/10.1002/bit.25133>)? NADH has a large change if I remember correctly.

- Is the dependence on glucose growth of all their solutions a weakness? The authors mention acetate growth as one reason why certain swaps are allowed – would any of their general findings miss aspects of cofactor choice because they are overly specific to glucose? Maybe adding an analysis of growth on acetate (or some other gluconeogenic substrate) as well would explain additional constraints on cofactor usage?

- The Github repository contains scripts to replicate the results of the paper. This is very welcome of course. The repository itself is quite flat (many functions in the top directory, and it's not immediately clear how to apply certain functions (in particular I was interested in the code to find sets of infeasible reaction constraints to remove). Additionally, documentation within the code functions themselves is quite scarce at a brief glance. Would the authors be open to creating a ReadTheDocs or similar documentation to explain better the use of their code? They seem to have written some quite useful code and good documentation would greatly improve its usefulness to the community.

Minor Comments

- A brief comparison of tFBA (employed by the Hatzimanikatis and Reed labs among others) to OptMDF would be welcome. To my understanding, they use identical constraints (the FBA constraints along with thermodynamic constraints), but tFBA uses the typical growth objective while OptMDF uses a linear objective which is the sum of Gibbs energies through the network, i.e. the total thermodynamic driving force. If I misunderstand the relationship between the two formulations or there are other differences that I missed, a clarification would be helpful.

- Could the authors please report the reactions that needed to be relaxed to obtain a feasible solution when applying the measured metabolite concentrations in Supplementary Text 3?

Reviewer #3 (Remarks to the Author):

Review of: Network-wide Thermodynamic Constraints Shape NAD(P)H Cofactor Specificity of Biochemical Reactions

Summary

The authors analyse why either NAD⁺ or NADP⁺ are used as electron acceptors across an E.coli genome-scale metabolic model and find that a great majority of cofactor swaps have a deleterious effect on thermodynamic efficiency indicating that the wild-type NAD(P)(H) specificities are, to a large extent, beneficial with respect to the overall thermodynamic driving force.

Major comments

- Does the β_i in eq 2 and 6 have the same meaning? It seems like the β_i in eq 6 is actually B. If so, it would be good to explain how eq 6 and 7 implement an approximation to the 2nd Law of Thermodynamics and what the limitations of this approximation are, if any.
- Please explain what is meant by thermodynamic efficiency early on in the manuscript.
- It is stated that thermodynamic data are “Gibbs free energies and metabolite concentrations”. However, metabolite concentrations are biochemical data as much as they are thermodynamic data, because they are the result of an evolutionary process, whereas “Gibbs free energies” are invariant with respect to evolution.
- Normally the NAD(H) pool is typically much more oxidized than the NADP(H) pool, if all NAD(H) are swapped for NADP(H) cofactors then would the NADP(H) pool be equally more oxidized than the NAD(H) pool in the same environment with the same metabolic objective? If they can be completely swapped, why is it optimal for the NAD(H) pool to be much more oxidized than the NADP(H) pool in the wild type?
- It might be worth considering, in future work, to what degree high thermodynamic driving forces are always optimal. For example, one could imagine that higher thermodynamic driving forces are also accompanied by greater release of heat and accrue a greater risk of thermally destabilising the

corresponding enzymes. Perhaps there is a trade off between enzymatic efficiency and thermal stability?

Minor comments

- “One conserved property of the metabolism” -> One conserved property of metabolism
- “As most important result, we find” -> As [the?] most important result, we find
- Opt(Sub)MDF is an unwieldy abbreviation for a concept. Can one not find a better (more readable), yet brief name for it?

"Network-wide Thermodynamic Constraints Shape NAD(P)H Cofactor Specificity of Biochemical Reactions"

Response to the comments of the reviewers

We thank the reviewers for their careful evaluation of our manuscript and useful suggestions. Below we outline how we addressed the comments.

Changes introduced in the manuscript are highlighted in that document.

Reviewer #1

The paper by Bekiaris and Klamt analyzes the NAD(P)H cofactor specificity of the metabolic network of *Escherichia coli*. The authors adapt an available genome-scale metabolic network for this purpose, by allowing swaps (changes) of preferred cofactors, and combine this network with available thermodynamic information for individual reactions (negative Gibbs free energy changes). Using the OptMDFpathway tool the authors predict a flux distribution maximizing the thermodynamic driving force of the network, for both the wild-type NAD(P)H specificities and different scenarios with perturbed specificities. The authors reach two very interesting main conclusions: (i) the wild-type NAD(P)H specificities give rise to a thermodynamic driving force close to the optimal, and (ii) random NAD(P)H specificities usually perform much worse than the wild-type NAD(P)H specificities. I find the methodology followed in this study exemplary and the conclusions very interesting and thought-provoking. My comments mostly concern the presentation and the interpretation of the results.

We thank the reviewer for the positive evaluation of our work.

1. I got a bit lost in the definition of the different forms of thermodynamic driving forces: max-min driving force, optimal max-min driving force, optimal max-min driving force for a network. A box or a figure intuitively explaining these concepts would help readers who are not already familiar with these concepts.

We revised the first paragraph of the section "Comparison of maximal thermodynamic driving forces for different cofactor specificity scenarios" in the Results section to better explain the different notions of thermodynamic driving forces. As suggested by the reviewer, we also included a figure to illustrate the concepts. Importantly, due to a comment of another reviewer, we also renamed OptMDF and OptSubMDF to MDF and SubMDF in the entire manuscript. As explained in the new paragraph, OptMDF is a generalization of the MDF (from a given pathway to an entire networks) but still relies on the main idea of MDF and the additional term Opt(Sub)MDF might then lead to confusion. We make clear that we use the generalized (network-wide) definition of MDF throughout the manuscript.

2. In the same way, the rationale of maximizing the max-min driving force is briefly explained in the Discussion, but the text would be easier to follow if the reason for optimizing this criterion was explained at an earlier stage already.

We agree and explain the rationale of using the MDF now already at the beginning of the Results section (at the end of the revised paragraph where the MDF and related concepts are introduced).

3. In comparison with MDFs, an OptMDF also contains information on fluxes and metabolite concentrations. This information is not really exploited in the text though (except for the computing the cofactor ratios). One question that I posed myself while reading the text is to which extent the predicted fluxes and concentrations for growth of wild-type *E. coli* using the model match those that have been observed experimentally? At present, only the predicted growth rate is compared with experimental data. And how does the predicted optimal thermodynamic driving force for wild-type specificities compare with the experimentally observed (global) driving force? The above comments concern the same point: how well do the predictions of the model, under the chosen objective criterion, capture the physiology of wild-type *E. coli*?

We agree that we should also present and discuss more concrete data regarding the obtained solutions for the wild-type specificity. Since discussing all 52 solutions (from all aerobic and anaerobic growth rates for which MDF and SubMDF were computed) would be impossible, we selected the MDF/SubMDF-optimal solutions for the wild-type for a high growth rate under aerobic and for a high growth rate under anaerobic conditions, which should reflect the reality best for non-limited growth of an *E. coli* wild-type strain. For these solutions we provide, for all reactions of the central metabolism and for all NAD(P)(H)-dependent reactions, results of (a) a flux variability without (loopless FVA) and with (TVA) consideration of thermodynamic constraints (these data also indicate which reactions are essential for the respective solution and whether this essentiality is due to stoichiometric or/and thermodynamic constraints); and (b) the maximal driving force of the reactions under the given optimal (Sub)MDF (indicating the thermodynamic bottlenecks). These results are presented in the new Supplementary Data 1.

While the predicted fluxes (and variabilities/essentialities) appear largely reasonable in light of biological knowledge, it is hard and possibly not meaningful to compare the predicted maximal driving forces with measured driving forces. First of all, there are only very few (metabolomic) datasets available, that would enable the calculation of a larger set of in vivo Gibbs free energies of reactions – simply because in most cases the concentration of at least one metabolite of a given reaction has/could not be measured: even in the cited paper of Bennett et al. (2009), from which we used the measured metabolite concentration data, only for 5 reactions a $\Delta G'$ could be provided (and this with assumptions regarding the intracellular phosphate concentration). Hence, only few in vivo data of Gibbs free energies of reactions are available at all. Second, the computed (Sub)MDF-optimal solutions represent extreme cases with maximal driving forces in the network. We do actually not expect that the cells reach these driving forces (at least not for all reactions as demanded by the MDF). For example, as now discussed in the Discussion section, the in vivo driving forces will be constrained by kinetic properties of the reaction mechanism and enzymes as well as by enzyme regulation, which are not considered in our MDF-based optimization. Nevertheless, analyzing the (Sub)MDF-optimal solutions is still meaningful in the sense that the reached (Sub)MDF indicates the thermodynamic flexibility of the system: a high (Sub)MDF value indicates that the metabolite concentrations can be chosen from a broader range to keep the flux distribution at higher driving forces or at least thermodynamically feasible (i.e., $\Delta G' < 0$ for all active reactions) - even if the theoretically feasible MDF itself is not reached.

We now also analyzed the feasible metabolite concentration ranges for the selected high growth rate for aerobic conditions (first under MDF-optimality, then under SubMDF-optimality), to investigate how well the calculated concentrations reflect vivo data. Accordingly, we performed a Concentration Variability Analysis (CVA) under standard concentration ranges with the optimal MDF or SubMDF as constraint and compared them with all measured metabolite concentrations ranges in the paper of Bennet et al. (2011). These data are also provided in the new Supplementary Data 1.

We first observed that many metabolite concentrations ranges had significant degrees of freedom but some metabolites show partially significant reductions in their upper or lower bounds including central metabolites such as ADP, ATP, NADH, NAD, 3-phosphoglycerate alanine, coenzyme-A, dihydroxyacetone-phosphate, glutamate, malate, or PEP. For both MDF as well as SubMDF we found that as many as 103 of the 107 measured concentration ranges overlap with the ranges predicted by CVA. Under MDF, the four metabolites lying outside of the computed range are ADP, L-glutamate, phosphoenolpyruvate and NADH; under SubMDF as target it are 2-oxoglutarate, 3-phospho-D-glycerate and again L-glutamate and NADH. For example, regarding NADH, the predicted upper bound lies below the measured range in both cases but is still relatively close to the experimentally determined range. The trend of NAD is correctly predicted to have a >250 times higher lower bound than NADH. Another example is glutamate: although the CVA predicted indeed an increased larger lower bound relative to the initial standard lower bound, the measured concentration was even higher than the standard upper bound of 0.02 M used in the model. Glutamate is known for being an osmotically important metabolite, an effect which was not considered herein.

We hope that these data and results added in Supplementary Data 1 provide more insights on the structure and properties of particular (Sub)MDF-optimal solutions under wild-type specificity.

4. In the discussion of reactions having a different optimal specificity depending on the growth conditions (e.g., growth on glucose or on acetate, p10), I was wondering whether *E. coli* possesses isoenzymes with different cofactor specificities that are differentially expressed depending on the growth condition.

We searched for reaction pairs in the iML1515 model that have a complementary NAD(P)H specificity. We found 33 such reaction pairs. Thereof, 22 are a result of promiscuous specificity of a single enzyme while only 11 pairs are catalyzed by different (iso)enzymes. However, almost all of them are located in anabolic pathways, which appear less relevant as targets for switching between different growth modes (they are always needed). We found only a single reaction (the malic enzyme reaction) in the central metabolism where the two specificities are catalyzed by two variants of the malic enzyme (SfcA, MaeB), which are indeed differently regulated. Hence we believe that differentially regulating isozymes is not a major mechanism used by *E. coli* to switch between different growth modes. We added the following paragraph in the section of cofactor swaps:

*“A potential mechanism for the cell to adapt the cofactor specificity of a reaction to varying growth conditions would be to use enzymes with different NAD(P)H specificities and to adjust their expression depending on the conditions. In the iML1515 model, we found 33 reaction pairs that can operate with both types of redox cofactors, of which 22 are promiscuous activities of the same enzyme and only 11 are catalyzed by isoforms with different cofactor specificity. However, there is only one such reaction pair in the central carbon metabolism, which is catalyzed by the malic enzyme decarboxylating malate to pyruvate. *E. coli* possess two variants of the malic enzyme (SfcA and MaeB), one (SfcA) preferring NAD⁺ and one (MaeB) working exclusively with NADP⁺, which are differently regulated [10]. Hence, the mechanism of switching between both cofactor*

specificities via isoforms of enzymes seems to be of lower relevance in E. coli, at least for central metabolic pathways.”

5. Another interesting prediction is that using three instead of two cofactor pairs does not contribute to higher thermodynamic driving forces. Is this a (logical?) consequence of the fact that a reaction has two directions and (which may come to the same thing) that cofactor recycling has a forward and a backward direction?

First of all, as was shown in the respective Figure, we actually found few growth rates where using three cofactors indeed led to higher (Sub)MDF values, although the gain in MDF is really low. Second, we mention that we included constraints in our calculations such that, in a given flux distribution, every reaction can only be used in one direction and only with one of two cofactor specificities (see Methods). Third, following a suggestion by Reviewer #2, we found that a third cofactor may indeed bring a real benefit if it has a different redox potential than the two redox couples NADH/NAD and NADPH/NADP (see our response to Reviewer #2).

Reviewer #2

In this work, the authors develop a thermodynamic framework to explore the theoretical basis for NADP/NAD choice and redundancy across the metabolic network of *E. coli*. This work closely mirrors a similar effort by Segre’s group published last year, which the authors discuss and cite, but the current effort goes farther in terms of a global thermodynamic analysis. The authors claim that the observed split of NAD and NADP dependent reactions in a real network (that of *E. coli*) is near optimal and their *in vivo* concentration ratio can be predicted from network-wide optimality, which is quite a satisfying result. The paper is written very clearly and the methods are simple to understand, utilizing the established MDF thermodynamic modeling framework. The authors include a sensitivity analysis of Gibbs energies (via sampling described in the supplement) and metabolite concentrations (via *in vivo* measured concentrations in the Supplement). My only comments are related to the depth of the analysis, which I hope the authors can expand.

We thank the reviewer for the positive evaluation of our work.

Major Comments

- It would be nice to see the single swap analysis expanded, specifically analyzing the relative impact of specific swaps to identify key reactions. Currently the finding seems to be that these swaps have little impact, quoting “our single-swap analysis showed either a negative or no effect on the respective MDF.” It is not clear how negative these MDF changes are, and it’s not clear that MDF offers a complete picture governing cofactor choice. For example, we never see cofactor swaps in most major enzymes like glycolysis across organisms, but the authors seem to indicate these are possible without repercussions. Presumably the concentrations would have to change drastically to accommodate swaps as well – would these shifts cause issues? Or would cofactor regeneration (flux considerations) become an issue somehow? What is the picture when this analysis incorporates real metabolite concentrations? The PDH example is interesting in this regard. This reaction is the entryway to the TCA cycle and one would think that its production of NADH

would be important to global energy production. Are there any side effects of swapping this reaction to NADPH specific, such as lower energy production capability?

First, we would like to point out that the quoted part actually states “*In the majority of cases*, our single-swap analysis showed either a negative or no effect on the respective MDF.” This means that we found quite a number of cases where we had a negative impact and some cases in which a single swap can have a positive contribution to the (Sub)MDF (as summarized in Table 2). Moreover, the impact (especially for negative effects) can be substantial. We acknowledge that we should provide absolute numbers indicating how much the (Sub)MDF can change upon cofactor swapping and therefore added maximal and average changes in the new Table 2.

Regarding the cofactor swaps in glycolysis we do only partially agree: the glyceraldehyde-3-phosphate dehydrogenase (GAPDH) reaction of the glycolysis often operates with NADP(H) instead of NAD(H) if it runs predominantly in direction of GAP (i.e. where the reduced cofactor is consumed), e.g. in organisms using the Calvin cycle. The case of the GAPDH reaction is now also discussed in the new section on the results with acetate as substrate (see response to Reviewer #2 below).

As suggested, we also ran the single cofactor swap analysis with the measured in vivo concentration ranges. We found that single swaps do not impact the MDF but can still have significant effects on the SubMDF: 27 reactions have a negative SubMDF influence of up to -9.4 kJ/mol with a cofactor swap, while only one swap (the isocitrate dehydrogenase) brings a positive impact of up to +6 kJ/mol. Thus, the results remain similar to the standard concentration ranges. The case of isocitrate dehydrogenase is also already discussed in the section “How many cofactor swaps are needed in the wild-type specificity to reach the theoretically maximal MDF?”

Finally, we do not think that a cofactor switch of PDH would necessarily lead to problems in the global energy production. First of all, NADH can still be produced in larger amounts in the TCA cycle, while swapping the specificity of PDH to NADPH avoids the necessary use of the reductive pentose phosphate pathway for NADPH production. Finally, even if NADPH would be produced in excess (relative to NADH), it could be converted to NADH through the transhydrogenase, which is thermodynamically favored because the NADP(H) pools is (also in the model) much more reduced than the one of NAD(H).

- In this vein, more analysis of individual reactions with this framework would be welcome. It's expected that due to smaller flux levels, most of the 238 nad/nadp reactions would not contribute much to the MDF and thus the key reactions determining MDF would come down to a much smaller number of relevant fluxes, is that not the case?

Generally, by definition, the MDF does not depend on the absolute flux levels. For the MDF, it only matters whether a reaction has a non-zero flux or not (and if it is non-zero then the deltaG of that reaction must at least reach the MDF, irrespective of its absolute flux level). However, the absolute flux will have an impact on the enzyme demand (and thus enzyme costs) which, however, is not considered in our MDF-based approach.

- Related to the above issue, what is the argument for why any enzyme with the thermodynamic flexibility to pick one cofactor or another does in fact pick the observed cofactor and not the other? A general set of rules governing this choice would be a nice result of this analysis. For example, why do some reactions have dual specificity? Is it lack of importance or some state specific

flexibility that is required related to the concentration ratios of the cofactors or necessity of regenerating them?

It is just the central goal of this work to show that the natural specificity enables high or even maximal MDF (being close to the optimal specificity with maximal MDF). However, it generally appears hard to formulate a general set of rules from our results. The reviewer is also right that with the MDF-based calculations we still have some degrees of freedom (i.e. changing the specificity of some reactions does not affect the MDF), as already discussed in the Discussion section. However, the overall likelihood of randomly finding a specificity with the high (Sub)MDF of the wild-type is very low as shown by the random specificity sampling. As discussed in the Discussion section, several alternative specificities may not be an option due to factors that cannot be covered by our MDF-based approach (e.g., regulation, kinetics, toxic metabolite concentration, metabolite channeling etc.).

- Further theoretical results with their framework would be welcome to add depth to their analysis. For example, is there any reason that NADH is the catabolic and NADPH is the anabolic cofactor, as opposed to vice versa? Any reason for those redox potentials in particular? What if a different molecule with a different potential was chosen? Any reason that certain pathways are chosen to regenerate NADPH?

First of all, the reviewer is right, one could expect that the model with the flexible specificity cannot distinguish between NAD(H) and NADP(H) and the roles of both cofactors might be reversed, i.e. NAD(H) pool highly reduced and NADP(H) more oxidized (this would not be expected in the wild-type specificity since the latter is fixed and optimized with the known in vivo concentration ratios). However, in our model, the choice between NAD(H) and NADP(H) in the flexible scenario is biased towards the wild-type situation, because we did not duplicate reactions using *both* NAD(H) and NADP(H) (as explained in the Methods section). For this reason, the transhydrogenases run with higher efficiency when electrons are transferred from NADPH towards NAD⁺ (the other direction is possible with one of the transhydrogenases but requires proton-motive force (coupling with transport of protons from periplasm to cytoplasm)). For this reason, when high growth rates are demanded, the model with free specificity prefers in the optimization the choice of NAD(H) for the more oxidized and NADP(H) for the more reduced pool of redox cofactors (as in the in vivo situation). To verify this, we performed test runs where the NAD(P)(H) cofactor specificities were completely swapped in the flexible specificity scenario (except for the reactions where both NAD(H) and NADP(H) are involved). Indeed, this leads to lowered (Sub)MDF values for the flexible scenario at high growth rates, while the (Sub)MDF did not change for medium or low growth rates (hence, for those growth rates, a total swap between NAD(H) and NADP(H) would not make a difference).

Finally, we thank the reviewer for the very useful suggestion to also consider the case where the redox couple of a cofactor has a different redox potential than NAD(P)(H). We added results from a simulation where we allowed a third cofactor pool to operate with higher or lower redox potential than NAD(P)(H). Indeed, (only) in this case, a third redox cofactor pool can bring a clear benefit compared to the case with only two redox cofactors.

- The authors try adding a third redox cofactor and show little effect on MDF. However, there are other redox cofactors present in real networks, (e.g. thioredoxin) – any idea on why?

First of all, as addressed in the context of the previous point, we tested the effect of using a third redox cofactor with a different redox potential and indeed found that this can further enhance the

(Sub)MDF. As highlighted in the extended section on usage of three cofactors, this shows that usage of more than two cofactors can indeed be beneficial. This may also explain existence of other redox cofactors in *E. coli* such as thioredoxin, although this electron carrier is involved in much less metabolic reactions and we speculate that its relevance is due to specialized electron transfer mechanisms (reduction of disulfide bonds in proteins). On the other hand, for organisms with complicated metabolic lifestyles (e.g. acetogens), a third widely used redox cofactor (e.g. ferredoxin; having a more negative redox potential) may pay off to balance complicated redox metabolisms. In the discussion section we now address this point by writing:

*“ On the other hand, analyzing a three-cofactor model, where a hypothetical third cofactor couple aside of NAD(H) and NADP(H) can be used, revealed that significant benefits for the (Sub)MDF are only possible if the redox couple of the third cofactor has a redox potential that is significantly different from NAD(P)H/NAD(P)⁺. This may explain the existence of other redox cofactors in *E. coli* such as thioredoxins. However, the latter is involved in much less metabolic reactions than NAD(P)(H) and it can be speculated that the relevance of electron carriers like thioredoxin lies more in enabling specialized electron transfer reactions (here: reduction of disulfide bonds of proteins). On the other hand, for organisms with complicated metabolic lifestyles, a third widely used redox cofactor may pay off. For example, carbon-fixing anaerobic acetogens (representatives of chemolithoautotrophic microorganisms) use, in addition to NAD(H) and NADP(H), ferredoxin (with a more negative redox potential) as a third major electron carrier in many redox reactions of the central metabolism giving them an extra degree of freedom to balance their complex redox metabolism.”*

- It's not clear what is learned from the growth rate perturbations. The MDF is reduced at the highest growth rates, suggesting that the optimal stoichiometric pathways place additional thermodynamic pressure on this system, if I understand correctly. This is expected but I do not understand what we learn from this exercise. The differences in MDF presumably are due to pathway choices at particular growth rates, so maybe additional investigation into these pathway choices and relationship to MDF would make this analysis richer.

If the cell grows with maximal growth rate, then there will be only few (possibly only a single) flux distribution(s) (pathway(s)) that enable this optimal growth. Due to this stoichiometric constraint, there is only little flexibility in choosing pathways that could enable higher (Sub)MDF because they would not reach the required growth rate. With decreasing growth rates, more and more pathways with higher thermodynamic potential become possible options which can then lead to an increase in the (Sub)MDF. We believe it is useful to see that the wild-type specificities are still superior over random specificities also for lower growth rates. The analysis also shows that under lower growth rates (or biomass yields) higher thermodynamic driving forces are possible (and quantifies this effect). However, as shown by the single swap analyses, conclusive results on specific reactions or pathways that are preferred under the different growth conditions (especially for lower growth rates) are hardly possible due to the mentioned remaining degrees of freedom.

- Results related to aerobic vs anaerobic growth are also relatively thin, as far as contributing to the overall picture of cofactor choice. Can the authors predict the metabolite shift from aerobic to anaerobic (e.g. using the data in McCloskey et al <https://doi.org/10.1002/bit.25133>)? NADH has a large change if I remember correctly.

First of all, we think that the results related to aerobic vs. anaerobic growth are very useful, as they demonstrate that the wild-type specificity is close to the MDF optimum under both conditions, which is not trivial.

Regarding the prediction of the metabolite concentration shift, we performed a Concentration Variability Analysis (CVA) under aerobic as well as anaerobic conditions (results of a CVA and comparison with measured concentrations for a selected (high) growth rate for aerobic and anaerobic conditions can be found in the new Supplementary Data 1). We found some smaller changes of the feasible concentration ranges but we were not able to identify particular or general trends. Hence, the few constraints used in our MDF-based approach may be limited in predicting absolute metabolite concentrations (which is not surprising since the latter will heavily depend on kinetic mechanisms and parameters not considered herein). On the other hand, our analysis of the quotient of NADH/NAD and NADPH/NADP shows that although absolute metabolite concentration (ranges) may be only weakly constrained in the optimal (MDF) states (as is the case for the NAD, NADH, NADP, NADPH metabolites) certain combinations thereof (here, certain ratios of reduced and oxidized cofactors) may indeed exhibit strong trends (in this case confirming that the NADP(H) pools must be much more reduced than NAD(H) in the MDF-optimal state). Similar relationships involving other metabolites may exist but may be complex (e.g. certain ratios) and hard to extract in a directed way (and could rather be subject of a follow-up study). Finally, the simulations with measured metabolite concentrations (posing much tighter constraints for the allowed metabolite concentration ranges) also demonstrated that the trends with respect to MDF-optimal states still hold.

We also looked at the data of the McCloskey paper cited by the reviewer and found that the NADH/NAD-ratio changed from $2.07 \text{ mM} / 5.84 \text{ mM} = 0.35$ (aerobic) to $9.71 \text{ mM} / 8 \text{ mM} = 1.2$ (anaerobic). While the ratio for aerobic conditions is significantly larger (less oxidized) than in the data we used from Bennett et al. (0.003), our results with the relative analysis of the reduced/oxidized ratio of NAD(P)H/NAD(P) in Figure 4 also indicates the possibility of a more reduced NAD(H) pool for higher growth rates, since its maximal reduction degree increases relative compared to NADP(H).

- Is the dependence on glucose growth of all their solutions a weakness? The authors mention acetate growth as one reason why certain swaps are allowed – would any of their general findings miss aspects of cofactor choice because they are overly specific to glucose? Maybe adding an analysis of growth on acetate (or some other gluconeogenic substrate) as well would explain additional constraints on cofactor usage?

We fully agree and thank the reviewer for this suggestion. Accordingly, we repeated all main analyses now also with acetate as substrate. The results can be found in a section added to the Results part. Strikingly, all main findings also hold true with acetate as sole substrate (note that only results for aerobic conditions are shown since *E. coli* cannot grow on acetate under anaerobic conditions).

- The Github repository contains scripts to replicate the results of the paper. This is very welcome of course. The repository itself is quite flat (many functions in the top directory, and it's not immediately clear how to apply certain functions (in particular I was interested in the code to find sets of infeasible reaction constraints to remove). Additionally, documentation within the code functions themselves is quite scarce at a brief glance. Would the authors be open to creating a ReadTheDocs or similar documentation to explain better the use of their code? They seem to have

written some quite useful code and good documentation would greatly improve its usefulness to the community.

We agree with the reviewer that the usability of the code could be improved. We therefore added many new docstring comments in all general and TCOSA-related modules and functions of the repository. Furthermore, the README was expanded and HTML-formatted documentations for all general functions (not specifically tied to the usage within this manuscript), since they are the most interesting ones for usage in other applications. In addition, the repository was cleaned up and unnecessary files were deleted.

Regarding the question about the handling of the bottleneck reactions, the respective routines can be found in the “cosa_load_model_data.py” script as described in its module comment. Furthermore, the underlying MILP can be found in the “optmdfpathway.py” script.

Minor Comments

- A brief comparison of tFBA (employed by the Hatzimanikatis and Reed labs among others) to OptMDF would be welcome. To my understanding, they use identical constraints (the FBA constraints along with thermodynamic constraints), but tFBA uses the typical growth objective while OptMDF uses a linear objective which is the sum of Gibbs energies through the network, i.e. the total thermodynamic driving force. If I misunderstand the relationship between the two formulations or there are other differences that I missed, a clarification would be helpful.

We added a short comparison between tFBA and MDF optimization problems in our Methods section, which should also answer the question of the reviewer:

“We note that the MILP optimization problem for the computation of MDF (eqs. (1)-(9)) overlaps to some extent with the MILP formulations of thermodynamics-based (metabolic) flux analysis (T(M)FA) and thermodynamics-based flux balance analysis (TFBA), which seek to analyze thermodynamically consistent flux distributions [30]. The key difference is that TFBA and T(M)FA explicitly demand that the Gibbs free energy of all reactions is negative, hence, that the reactions obey the second law of thermodynamics. Within these constraints, TFBA is used to optimize certain fluxes (e.g. growth rate as usual for FBA), metabolite concentrations or particular Gibbs free energies. In our MDF formulation, the objective function (9) maximizes the MDF B , i.e. it maximizes the minimal driving force of all active reactions. The Gibbs free energies are hidden behind the driving forces f_i and the MDF B (the latter capturing the smallest of all relevant driving forces). In most calculations, we did not explicitly demand that $B > 0$ (which would imply $\Delta_r G'_i < 0$ for all reactions) but one can easily check feasibility of the found solutions by verifying that $B > 0$. If $B \leq 0$ then the absolute value of B indicates the distance of the solution from thermodynamic feasibility.”

From this, please note that our (Opt)MDF-based formulations does not optimize the sum of all driving forces, instead, it maximizes the minimal driving force of all active reactions.

- Could the authors please report the reactions that needed to be relaxed to obtain a feasible solution when applying the measured metabolite concentrations in Supplementary Text 3?

We now list these thermodynamic bottleneck reactions in the added Supplementary Table 2.

Reviewer #3

Summary

The authors analyse why either NAD⁺ or NADP⁺ are used as electron acceptors across an E.coli genome-scale metabolic model and find that a great majority of cofactor swaps have a deleterious effect on thermodynamic efficiency indicating that the wild-type NAD(P)(H) specificities are, to a large extent, beneficial with respect to the overall thermodynamic driving force.

Major comments

- Does the β_i in eq 2 and 6 have the same meaning? It seems like the β_i in eq 6 is actually B. If so, it would be good to explain how eq 6 and 7 implement an approximation to the 2nd Law of Thermodynamics and what the limitations of this approximation are, if any.

The β_i in equations 2 and 6 do indeed have the same meaning (upper bound of flux of reaction i). B represents the MDF (to be calculated by objective function (9)) under the given settings, and has no relation to the (maximal) flux of a reaction. Instead, eqs (4), (6), and (7) ensure that the driving force of all active reactions reach at least the value of B (the latter representing the minimal driving force of all active reactions). We do not explicitly demand that B (and thus the driving forces of all active reactions) must lie above 0. Instead, when computing the MDF (B) we can easily assess whether a found solution is thermodynamically feasible or not (if $B > 0$ then $\Delta_r G' < 0$ for all active reactions).

We now explain this more explicitly in the Methods section (after eq. (9)), where we also clarify the relationships with TFBA and TMFA (as demanded by another reviewer).

- Please explain what is meant by thermodynamic efficiency early on in the manuscript.

We understand that this term is imprecise and therefore changed the formulation to “thermodynamic potentials”.

- It is stated that thermodynamic data are “Gibbs free energies and metabolite concentrations”. However, metabolite concentrations are biochemical data as much as they are thermodynamic data, because they are the result of an evolutionary process, whereas “Gibbs free energies” are invariant with respect to evolution.

We agree with this more precise differentiation of biochemical and thermodynamic data and changed the text accordingly to avoid misunderstandings.

- Normally the NAD(H) pool is typically much more oxidized than the NADP(H) pool, if all NAD(H) are swapped for NADP(H) cofactors then would the NADP(H) pool be equally more oxidized than the NAD(H) pool in the same environment with the same metabolic objective? If they can be completely swapped, why is it optimal for the NAD(H) pool to be much more oxidized than the NADP(H) pool in the wild type?

First of all, the reviewer is right, one could expect that the model with free specificity cannot distinguish between NAD(H) and NADP(H) and the roles of both cofactors might be reversed (NAD(H) pool highly reduced and NADP(H) more oxidized). However, in our model, the choice

between NAD(H) and NADP(H) is biased towards the wild-type situation, because we did not duplicate reactions using *both* NAD(H) and NADP(H) (as explained in the Methods section). For this reason, the transhydrogenases run with higher efficiency when electrons are transferred from NADPH towards NAD⁺ (the other direction is possible with one of the transhydrogenases but requires proton-motive force (coupling with transport of protons from periplasm to cytoplasm)). For this reason, when high growth rates are demanded, the model with free specificity prefers in the optimization the choice of NAD(H) for the more oxidized and NADP(H) for the more reduced pool of redox cofactors (as in the *in vivo* situation). To verify this, we performed test runs where the NAD(P)(H)cofactor specificities were completely swapped in the flexible specificity scenario (except for the reactions where both NAD(H) and NADP(H) are involved). Indeed, this leads to lowered (Sub)MDF values for the flexible scenario at high growth rates, while the (Sub)MDF did not change for medium or low growth rates (hence, for those growth rates, a total swap between NAD(H) and NADP(H) would not make a difference).

- It might be worth considering, in future work, to what degree high thermodynamic driving forces are always optimal. For example, one could imagine that higher thermodynamic driving forces are also accompanied by greater release of heat and accrue a greater risk of thermally destabilising the corresponding enzymes. Perhaps there is a trade off between enzymatic efficiency and thermal stability?

We thank the reviewer for this suggestion. Indeed, in a future work, we could think further about how states with high thermodynamic driving forces affect other physical and biological factors, but we feel that it would go beyond the scope of this paper.

Minor comments

- “One conserved property of the metabolism” -> One conserved property of metabolism

Thanks, fixed.

- “As most important result, we find” -> As [the?] most important result, we find

Thanks, fixed.

- Opt(Sub)MDF is an unwieldy abbreviation for a concept. Can one not find a better (more readable), yet brief name for it?

We agree that OptMDF/OptSubMDF might be a bit clumsy. Based on this comment and another suggestion given by Reviewer #1 we revised the first paragraph of the section “Comparison of maximal thermodynamic driving forces for different cofactor specificity scenarios” in the Results section to better explain the different notions of thermodynamic driving forces. As suggested by reviewer #1, we also included a figure to illustrate the concepts. Due to the comment of this reviewer (#3), we also renamed OptMDF to MDF and OptSubMDF to SubMDF in the entire manuscript. As explained in the new paragraph, OptMDF can be seen a generalization of the MDF (from a given pathway to networks) but still relies on the main idea of MDF and the additional term Opt(Sub)MDF might then lead to confusion. We make clear that we use the generalized (network-wide) definition of MDF (and SubMDF) throughout the manuscript. We hope that these terms are easier accessible.

REVIEWERS' COMMENTS

Reviewer #1 (Remarks to the Author):

The authors have convincingly addressed all points that I raised in my previous review.

Reviewer #2 (Remarks to the Author):

The authors have sufficiently addressed my comments. I am happy to see expanded analysis and discuss of their results. The improvements to their code repository are welcome as well.

I am enthusiastic about the findings of the paper and the continued growth of thermodynamic tools applied to answer fundamental questions about metabolism. I have no further issues.